# HUMANVBENCH: PROBING HUMAN-CENTRIC VIDEO UNDERSTANDING IN MLLMS WITH AUTOMATICALLY SYNTHESIZED BENCHMARKS

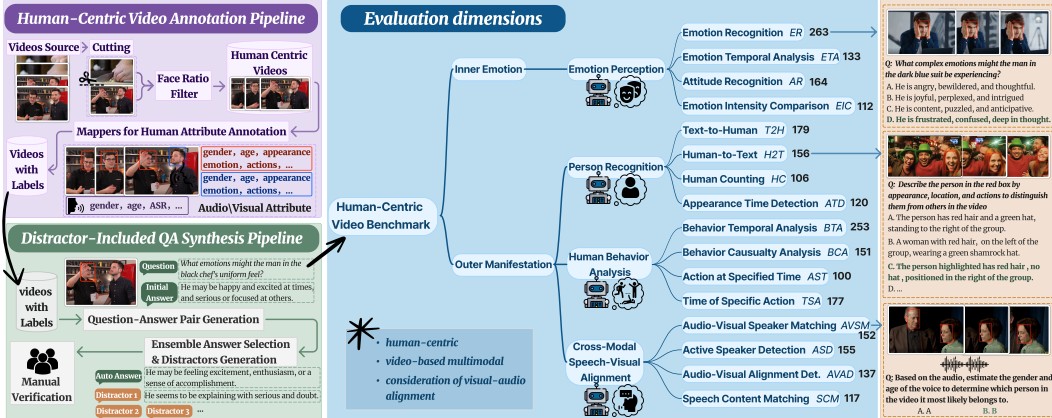

Figure 1: HUMANVBENCH is a benchmark for human-centric video understanding that features 16 fine-grained tasks, each denoted by its acronym and the number of included QA instances (middle blue box). It is synthesized with minimal human intervention by using a Video Annotation Pipeline (upper left, purple box) and a Distractor-Included QA Synthesis Pipeline (lower left, green box).

## ABSTRACT

Evaluating the nuanced human-centric video understanding capabilities of Multimodal Large Language Models (MLLMs) remains a great challenge, as existing benchmarks often overlook the intricacies of emotion, behavior, and cross-modal alignment. We introduce HUMANVBENCH, a comprehensive video benchmark designed to rigorously probe these capabilities across 16 fine-grained tasks. A cornerstone of our work is a novel and scalable benchmark construction methodology, featuring two automated pipelines that synthesize high-quality video annotations and challenging multiple-choice questions with minimal human labor. By leveraging state-of-the-art models for annotation and systematically converting model-induced errors into plausible distractors, our framework provides a generalizable "machine" for creating nuanced evaluation suites. Our extensive evaluation of 27 leading MLLMs on HUMANVBENCH reveals critical deficiencies, particularly in perceiving subtle emotions and aligning speech with visual cues, with even top proprietary models falling short of human performance. We open-source HUMANVBENCH and our synthesis pipelines to catalyze the development of more socially intelligent and capable video MLLMs.

## 1 INTRODUCTION

In recent years, Multimodal Large Language Models have emerged as pivotal advancements, extending traditional language models to process diverse modalities such as text, images, and videos (Qin et al., 2024). Video-oriented MLLMs, in particular, have drawn growing interest for their potential to interpret video content in ways closely aligned with human perception (Tang et al., 2025). However, the extent to which these models truly achieve human-like understanding, especially in complex human-centric scenarios, remains an open question.

Human-centric scenes in videos naturally attract attention due to the emphasis on individuals' emotions, actions, and verbal interactions, necessitating effective comprehension by video understanding models. Despite advances in this field, existing benchmarks often fall short in rigorously assessing the nuanced understanding of human emotions and behaviors. Current evaluations predominantly focus on general content comprehension, object recognition, and action detection, often neglecting subtle intricacies such as emotional insight and behavioral analysis. Furthermore, synchronizing speech with visual elements remains a substantial challenge; unlike humans, who effortlessly discern mismatches between audio and visual cues, computational models often struggle with tasks like identifying speakers and aligning speech with corresponding lip movements. Bridging these fine-grained gaps demands not only new evaluation datasets, but a more scalable and systematic paradigm for their creation.

To address these limitations, we introduce a novel and scalable framework for automated video benchmark creation. This framework is powered by two core pipelines: 1) the Human-Centric Video Annotation Pipeline, which leverages over twenty state-of-the-art (SOTA) data processing operators to produce dense, multi-modal, and automated video annotations; and 2) the Distractor-Included QA Synthesis Pipeline, which generates high-quality multiple-choice questions with semantically plausible distractors. As a powerful instantiation of our framework, we present HumanVBench, a pioneering benchmark featuring 16 fine-grained QA tasks specifically tailored for human-centric analysis in video MLLMs. These tasks are organized around two core dimensions: inner emotion and outer manifestation, as depicted in Figure 1.

A key innovation of our framework lies in its model-driven design. Unlike conventional benchmarks that rely on laborious human annotation, our approach automates the multi-modal, fine-grained annotation process by integrating diverse task-specific algorithms and models. The Distractor-Included QA Synthesis Pipeline further ensures the challenging nature of the questions by adeptly leveraging multi-model ensembles and their common failure modes to generate semantically deceptive distractors. The strategy transforms the human role from manual creation to efficient quality review, significantly boosting both construction efficiency and scalability. Moreover, our method is easily adaptable to "in-the-wild" video data, enabling the creation of benchmarks that extend beyond controlled or domain-specific environments.

Through HUMANVBENCH, we conduct a comprehensive evaluation of 27 state-of-the-art video MLLMs, covering both open-source and commercial models. Our results reveal notable gaps between current model capabilities and human-level video understanding, particularly in nuanced emotion perception. While the proprietary models demonstrate closer human-like accuracy, open-source models frequently misclassify emotions due to temporal noise, underscoring the need for further architectural improvements and refined datasets.

In summary, our contributions are as follows:

• We introduce HUMANVBENCH, a novel video benchmark for MLLMs that emphasizes fine-grained human comprehension in videos, focusing on emotion perception, person recognition, behavioral analysis, and speech-visual alignment.

• We propose a novel and scalable framework for automated video benchmark construction, featuring two advanced pipelines for multi-modal annotation and distractor-aware QA synthesis. Our framework is generalizable and significantly reduces manual annotation effort.

• Our comprehensive evaluation of 27 SOTA video MLLMs offers key insights, facilitating in-depth discussions regarding their performance, strengths, and areas for enhancement.

• We release our benchmark, including data, evaluation, and synthesis codes at anonymous link , to foster further evolution of future human-centric video analysis systems.

## 2 RELATED WORKS

**Multimodal Large Language Models.** The remarkable progress in Large Language Models (LLMs) has sparked extensive research into merging language comprehension with visual and auditory information, thereby expediting the advancement of multimodal models. Within this domain, image MLLMs amalgamate visual and linguistic data to enhance image interpretation and cross-modal reasoning (Chen et al., 2024f; Jiao et al., 2025a; Liu et al., 2024). Video MLLMs extend

these capabilities by incorporating temporal sequences for dynamic video analysis (Li et al., 2023; Maaz et al., 2024; Chen et al., 2025; Bai et al., 2025). Furthermore, generalist MLLMs can process a plethora of inputs, including images, video, and audio, thus improving adaptive task performance across various modalities (Han et al., 2024; Girdhar et al., 2023; Zhao et al., 2023). Despite these advancements, rigorous evaluation of video MLLMs on human-centric video understanding tasks remains an unaddressed challenge, which this work aims to address.

**Video Benchmarks for MLLMs.** Recent years have seen the emergence of diverse benchmarks for assessing MLLMs, with a particular emphasis on video-based evaluations. Representative efforts such as Video-MME (Fu et al., 2025), MVBench (Li et al., 2024c), and LvBench (Zhang et al., 2025b) are primarily designed to measure general video understanding, covering capabilities like temporal reasoning and broad scene comprehension. Although human elements are present, they are dispersed across general tasks without structured or fine-grained evaluation, limiting their diagnostic power in this domain. To more precisely evaluate models' understanding of human activities, several human-focused benchmarks have been introduced. For instance, ActivityNet-QA (Xu et al., 2016) focuses on object and human activity recognition. HumanBench (Tang et al., 2023) amalgamates multiple datasets to create benchmarks for tasks like pose estimation, pedestrian attribute recognition, and crowd assessment. HERM (Li et al., 2024b) is a human-centric image benchmark focusing on people, poses, actions, and interactions. MotionBench (Hong et al., 2025) tests models' perception of motion details with questions and options created by human annotators.

**Our Position.** Compared to existing benchmarks, ours stands out with both innovative task design and unique construction methodologies. In terms of evaluation dimensions, unlike general-purpose video benchmarks, ours features 16 fine-grained tasks and around 2.5k human-tailored queries. Compared with human-related benchmarks, ours goes further in two ways: unlike HERM, it incorporates complex scenarios with temporal dynamics for richer human video interpretation; and beyond ActivityNet-QA, HumanBench, and MotionBench, it captures nuanced emotional comprehension and evaluates cross-modal alignment between visual and speech modalities. Moreover, HUMANVBENCH boasts a pioneering construction approach, deriving from raw, uncurated video content and employing over 20 advanced processing operators for meticulous video character annotation and automatized multiple-choice question formulation. Human intervention is minimized to verify the quality of well-structured QAs.

## 3 THE PROPOSED HUMANVBENCH

### 3.1 TASK DESIGN AND DEFINITION

Human observers naturally focus on individuals in videos, attending to their appearance, emotions, and behaviors. This intrinsic focus underpins our design of 16 fine-grained, human-centric tasks, aimed at evaluating MLLMs' ability to mimic human-like perception in video analysis (Figure 1). Each task definitions and examples are provided in Appendix G. The tasks can be grouped into two categories based on content observability: *Inner Emotion* and *Outer Manifestation*.

Inner emotions are often less observable in real-world videos, requiring abstract skills to interpret subtle facial expressions, body language, and verbal cues. The tasks in this category, which we term **Emotion Perception**, evaluate MLLMs' ability to detect and interpret these emotional nuances. Specifically, this category includes: *Emotion Recognition* (ER) identifies the most fitting emotional description for an individual. *Emotion Temporal Analysis* (ETA) tracks emotional changes of a person over time. *Attitude Recognition* (AT) assesses the inferred attitude (positive, negative, or neutral) of individuals toward specific events or entities. *Emotion Intensity Comparison* (EIC) differentiates and quantifies the emotional intensity of various individuals.

In contrast to inner emotions, outer manifestations deal with more tangible aspects such as identifying individual(s), causality reasoning, and synchronization of video elements like speech or singing. Guided by these aspects of human observation, we formulated three task categories:

**Person Recognition** evaluates the model's capability to identify a particular person within a complex scene, akin to "person-finding": *Text-to-Human* (T2H) identifies a person based on a textual description. *Human-to-Text* (H2T) assesses the accuracy of a textual explanation attributed to a target person. *Human Counting* (HC) detects, tracks, and counts distinct individuals. *Appearance Time Detection* (ATD) identifies the specific time a specified individual appears.

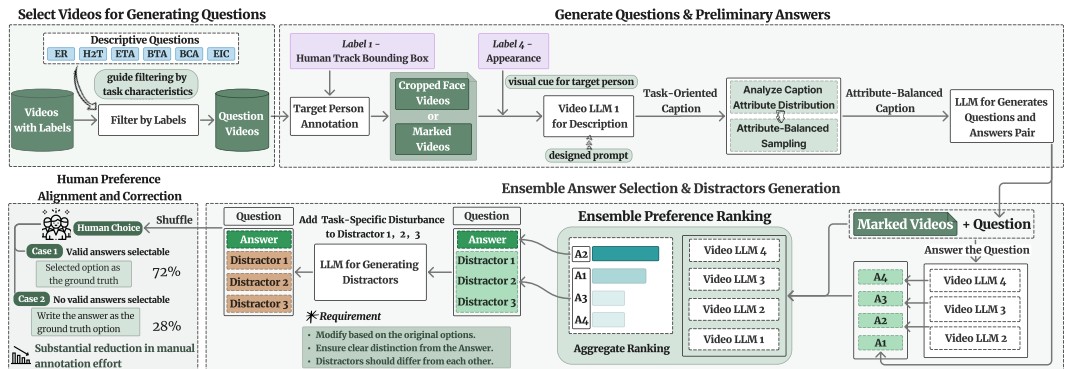

Figure 2: The Human-Centric Video Annotation Pipeline involves obtaining videos featuring people and annotating both visual and auditory information as well as overall event atmospheres.

Figure 3: The Distractor-Included QA Synthesis Pipeline facilitates four steps: selecting "question videos", generating Questions, ensemble-based options generation with model-derived errors as distractors, and manually verifying multiple-choice questions.

**Behavior Analysis** scrutinizes a model's ability to understand and analyze individual behaviors, with four tasks: *Behavior Temporal Analysis* (BTA) tracks changes in a specified individual's behaviors over time. *Behavior Causality Analysis* (BCA) examines causal reasoning of behaviors in the video context. *Action at Specified Time* (AST) identifies the precise actions occurring at a given time. *Time of Specific Action* (TSA) detects the exact moments when specific actions occur.

**Speech-Visual Alignment** is a critical cross-modal category designed to test capabilities like speaker identification and lip-syncing. This category includes: *Audio-Visual Speaker Matching* (AVSM) correlates audio features to identify individuals and their visual appearance (gender, age). *Active Speaker Detection* (ASD) identifies the individual currently speaking by integrating visual and audio inputs. *Audio-Visual Alignment Detection* (AVAD) detects synchronization points and coherence between lip movements and audio. *Speech Content Matching* (SCM) analyses spoken content against text, evaluating transcription or lip-reading capabilities.

Together, these tasks offer a comprehensive assessment of MLLMs' emotion, identity, behavior, and speech-visual alignment understanding, driving towards human-like video comprehension.

## 3.2 HUMAN-CENTRIC VIDEO ANNOTATION PIPELINE

Creating task-specific questions for the aforementioned tasks hinges on extensive human-centric annotations within videos. Our video annotation pipeline, illustrated in Figure 2, emphasizes multi-modal, granular annotation pathways, supported by the Data-Juicer framework (Chen et al., 2024a;c). While benefiting from existing operators, we've also developed innovative operators to enhance the open-source community. Detailed annotations and examples are in the Appendix H.

**Collecting Videos Containing People.** We sourced copyright-free videos from Pexels and movies in the MF2 benchmark (Zaranis et al., 2025) (a benchmark assessing full-movie comprehension) under the Public Domain 1.0 license. Importantly, we used only the raw visual data

without any pre-existing captions or metadata. All annotations were generated internally by our pipelines, thereby preventing risks of data leakage. We begin by splitting each video into scenes with `video_split_by_scene_mapper` to facilitate subsequent accurate human tracking. The resulting clips are then filtered by attributes such as duration and optical flow, after which `video_face_ratio_filter` is applied to ensure that faces are visible in most frames. This guarantees that the selected videos are primarily human-centric and ready for further annotation.

**Human-Centric Video Annotation Pipeline.** We design a set of operators to extract and enrich human-centered video information. First, `video_human_tracks_extraction_mapper` links detected faces and bodies across consecutive frames using overlap thresholds, producing reliable person tracks and an approximate count of individuals in a shot. These tracks support subsequent tasks such as person highlighting and appearance/action descriptions.

From the tracks, `human_demographics_mapper` extracts face crops and applies attribute models to infer demographic labels (e.g., age, gender, race). Flexible bounding box crops (full-body or face-only) further allow generating individual-focused clips. Based on this, `video_human_description_mapper` captures appearance and posture using an MLLM, while `video_facial_description_mapper` focuses on facial expressions and changes. This ensures descriptions are unaffected by background or other people, while preserving facial detail.

For audio, `audio_tagging_mapper` first classifies sound types. If speech is present, additional operators are invoked: `active_speaker_detection_mapper` fuses audio-visual cues to localize speakers, `asr_mapper` transcribes speech, `speech_emotion_recognition_mapper` detects emotions, and `voice_demographics_mapper` profiles vocal attributes (e.g., gender, age).

Finally, `video_description_mapper` summarizes overall atmosphere and narrative using MLLMs. These multimodal annotations enable automated task construction. For example, in the *Audio-Visual Speaker Matching* task, videos are selected where exactly one individual's visual demographics (*Label-3*) match the audio demographics (*Label-10*), while other individuals' features mismatch. The matched individual serves as ground truth, with IDs assigned via *Label-1* tracking. Implementation details for each operator and full task construction appear in Appendices H and I.

### 3.3 DISTRACTOR-INCLUDED QA GENERATION PIPELINE

For tasks with exact answers (e.g., restricted categories, numbers, or letters), questions, correct answers, and distractors are constructed using tailored templates and annotations from Section 3.2. For open-ended questions, we developed a pipeline to generate questions, answers, and distractors, applied to six tasks in HUMANVBENCH: *Emotion Recognition, Emotion Temporal Analysis, Emotion Intensity Comparison, Human-to-Text, Behavior Temporal Analysis*, and *Behavior Causality Analysis*. The pipeline (Figure 3) consists of four stages:

**Selecting Videos.** Videos are filtered by task-specific criteria. For example, *Behavior Temporal Analysis* requires longer clips, while *Human-to-Text* requires at least two people.

**Question and Preliminary Answer Synthesis.** Firstly, target persons are localized using bounding box tracks. For facial emotion-focused tasks, we reconstruct the video with cropped face regions; for others, the target person is highlighted with a red bounding box to create "marked videos". These marked videos are fed into a Video-MLLM with prompt designed for the specific task, so that the model produces targeted descriptions such as emotional states or behavior summaries. Since Video-MLLMs may not always focus on the highlighted individual, we incorporate a visual cue from *Label-4* (Figure 2) into the prompt. The generated descriptions are then analyzed to extract task-relevant attributes, and their distributions are adjusted to ensure balance (e.g., maintaining similar proportions of positive and negative emotions). Based on these captions, an LLM (GPT-4 in our workflow) is used to generate questions and preliminary answers.

**Ensemble Answer Selection and Distractors Generation.** To mitigate errors from single-model answers, we adopt an ensemble strategy. Multiple MLLMs (Gemini (Team et al., 2023), VideoL-LaMA3 (Zhang et al., 2025a), ShareGPT4Video (Chen et al., 2024e)) produce candidate answers, which are ranked via preference voting. The top choice is taken as the correct answer, while the others are converted into distractors: if semantically distinct (valid answer component), they are retained; otherwise an LLM (GPT-4 in our workflow) introduces task-specific perturbations. Retaining these erroneous answers allows the distractors to reflect typical model mistakes, thereby ensuring

plausibility and distractive power rather than being arbitrarily fabricated, which increases the overall difficulty of the questions.

**Manual Verification and Correction** Given the annotation models' limitations, errors may occur in the multiple-choice questions. To mitigate this, we adopt a human verification stage: all candidate options are shuffled, and annotators are instructed to select the correct one; if none of the options is fully accurate, they directly write the correct answer, which is then incorporated as the ground truth.

The Distractor-Included QA Synthesis Pipeline handles large-scale question and distractor generation, while humans serve as final arbiters of correctness. This substantially reduces manual workload—annotators only needed to rewrite the correct answer in about 25% of cases, while in the remaining 75% they confirmed one of the generated options (see Section 4.4). To further ensure reliability, we conducted an inter-annotator agreement (IAA) study on 240 randomly sampled questions from the final verified questions, where two independent annotators achieved a Cohen's Kappa of 0.8833, demonstrating the reliability of our benchmark.

### 3.4 POST-PROCESSING

To address the prevalent issue of answer leakage in multimodal evaluation datasets, we adopt the approach proposed in (Chen et al., 2024d). Specifically, we test models without visual input and remove frequently correct QAs ($\tilde{6}$%), ensuring random accuracy. This preserves visual relevance while mitigating leakage risks. In total, HUMANVBENCH contains 2475 problem instances; per-task counts are shown in Figure 1, with additional statistics in Appendix C.

## 4 EVALUATION AND INSIGHTS

### 4.1 EXPERIMENTAL SETTINGS

We evaluate 27 of the most popular video MLLMs, including visual-only MLLMs from the Qwen-VL series (Bai et al., 2025), InternVL series (Chen et al., 2025), and VideoLLaMA3, as well as audio-visual MLLMs like Qwen2.5-Omni (Xu et al., 2025) and Video-LLaMA series (Zhang et al., 2023; Cheng et al., 2024). We also evaluated commercial models from the GPT-4o, GPT-5, and Gemini series. All tasks were cast as multiple-choice questions (N-choose-1, with variable N), with human and random guesses as baselines. For more implementation details and the full results of all 16 tasks on 27 MLLMs , please refer to Appendix F and D.

### 4.2 COMPREHENSIVE EVALUATION OF MLLMs ON HUMANVBENCH

Table 1 summarizes the performance evaluation results for our benchmark, revealing several key insights from analyzing vision-only tasks (emotion perception, person recognition, and behavior analysis tasks) and cross-modal tasks (speech-visual alignment tasks).

#### 4.2.1 PERFORMANCE IN VISION-ONLY TASKS

**Open-Source Video-MLLMs:** Overall, current models still exhibit a clear gap from human-level performance, especially in Emotion Perception tasks, underscoring their limited ability to capture emotional nuances. Nevertheless, among open-source models, VideoLLaMA3 leads in video human understanding, and achieves a modest 54.7% mean accuracy across 12 vision-only tasks. Although it still lags behind human performance, it has surpassed GPT-4o and approaches other commercial counterparts, highlighting the immense potential of open-source models.

**Proprietary MLLMs:** Overall, with the exception of GPT-4o, proprietary models substantially outperform open-source ones across tasks, though their capability in emotion perception remains relatively weak, underscoring the need for improvement in nuanced emotion understanding; notably, Gemini-2.5-Pro stands far ahead, achieving near-human proficiency in person recognition, whereas GPT-4o performs unexpectedly poorly; its subpar performance in the Emotion Understanding category and in specific Person Recognition tasks (T2H and ATD) results in its average performance being lower than that of several leading open-source models.

Table 1: Summary of performance on HUMANVBENCH. We report average accuracy (%) across four main categories. Full results for all 24 models and 16 tasks are in Appendix D. The best overall results are shown in bold, while the best open-source model results are underlined "-" means the model recognizes its lack of required capabilities for the task and thus refuses to answer.

| Model | Mod. | Emotion 4 tasks | Person Recog. 4 tasks | Behavior 4 tasks | Avg. 12 tasks | Speech-Visual Alignment | | | | | Overall Avg. 16 tasks |
|---|---|---|---|---|---|---|---|---|---|---|---|
| | | | | | | AVSM | ASD | AVAD | SCM | 4 tasks | |
| Random Guess | | 24.4 | 25.2 | 22.9 | 24.2 | 42.8 | 23.6 | 33.3 | 25.0 | 31.2 | 25.9 |
| Qwen-VL2.5 (7B) | V | 39.8 | 61.3 | 48.5 | 49.9 | 71.1 | 61.3 | 33.6 | 18.8 | 46.2 | 49.0 |
| InternVL2.5 (7B) | V | 41.8 | 57.7 | 53.4 | 51.0 | 65.1 | 61.3 | 32.1 | 15.4 | 43.5 | 49.1 |
| VideoLLaMA3 (7B) | V | 39.7 | 68.5 | 55.8 | 54.7 | 64.6 | 63.1 | 34.4 | 17.9 | 45.0 | 52.3 |
| VideoLLaMA (7B) | V+A | 21.1 | 23.8 | 25.7 | 23.5 | 40.1 | 26.6 | 33.1 | 26.2 | 31.5 | 25.5 |
| VideoLLaMA2.1 (7B) | V+A | 36.5 | 36.4 | 43.6 | 38.8 | 44.0 | 31.6 | 32.1 | 23.7 | 32.9 | 37.4 |
| Qwen2.5-Omni (7B) | V+A | 35.5 | 44.5 | 38.3 | 39.4 | 71.1 | 48.4 | 27.0 | 71.8 | 54.6 | 43.2 |
| GPT-4o | V | 33.6 | 50.9 | 62.1 | 48.9 | - | - | - | - | - | - |
| GPT-5 | V | 46.8 | 69.5 | 67.3 | 61.2 | - | - | - | - | - | - |
| Gemini-1.5-Pro | V+A | 50.9 | 71.4 | 60.7 | 61.0 | 90.1 | 76.8 | 66.4 | 84.6 | 79.5 | 65.6 |
| Gemini-2.5-flash | V+A | **53.0** | 75.7 | 64.9 | 64.5 | **98.7** | **80.6** | 57.7 | **99.1** | 84.0 | 64.5 |
| Gemini-2.5-Pro | V+A | 52.9 | **83.5** | **70.7** | **69.0** | 96.7 | 78.7 | **72.3** | 98.3 | **86.5** | **73.4** |
| Human (Graduate) | | 84.6 | 88.5 | 87.0 | 86.7 | 96.0 | 96.1 | 87.0 | 98.3 | 94.4 | 88.6 |

### 4.2.2 PERFORMANCE IN SPEECH-VISUAL ALIGNMENT TASKS

**Lip-Reading Ability of Visual-Only MLLMs** For AVSM and ASD tasks, many videos include dubbing or multi-speaker conversations with single-speaker audio, making lip movements alone insufficient for accurate responses. Therefore, when answering these two tasks, visual-only models essentially degrade to speech action recognition but can still get some questions correct, with QwenVL2.5 outperforming other models. However, for the AVAD and SCM task, almost all models perform at a random level, indicating that current models lack precise lip-reading (lip translation) ability. Future models could focus on enhancing lip-reading capabilities.

**Audio-Video MLLMs:** Although most open-source MLLMs can process both audio and visual inputs, they—except for Qwen2.5-Omni—perform near-randomly on Speech-Visual Alignment tasks, particularly AVSM and ASD, where they even lag behind visual-only models. This deficiency may stem from two key factors: a limited ability to visually interpret intricate lip movements (as further detailed in Appendix Section E) and a general lack of a suitable temporal processing mechanism to facilitate fine-grained speech-to-lip alignment, further exacerbated by scarce datasets providing audio-visual lexical mappings. Consequently, while dedicated ASR models can effectively handle SCM, most open-source video-MLLMs struggle acutely with correlating speech to lip movements. In contrast, the Gemini series shows exceptional cross-modal alignment, performing robustly even on AVAD, a task no other models can handle.

### 4.3 DISSECTING FINE-GRAINED INSIGHTS

### 4.3.1 DISCUSSION ON SPEAKER EMOTION RECOGNITION

A detailed analysis reveals that models frequently misattribute speaker emotions as "surprise" or "shock". This issue primarily stems from frame sampling processes, where frames depicting "mouth-opening" motions are misclassified. Figure 4 illustrates this phenomenon using the commonly adopted eight-frame fixed sampling approach (Lin et al., 2024a; Zhang et al., 2024), which introduces temporal noise and leads to erroneous emotional judgments. The speaker-centric evaluations presented in Table 2 further confirm a consistent decline in accuracy across all models, underscoring the inherent challenges of this task for video MLLMs.

### 4.3.2 IMPACT OF TIMESTAMP INCLUSION ON TIME-SPECIFIC TASKS

Our findings show that many open-source video-MLLMs struggle with time-specific tasks due to their inability to correlate video events with a timeline (Appendix Table D). While top performers like VideoLLaMA3 and LLaVA-Video natively incorporate textual timestamps, we explored this capability by adding timestamps to models without native support. As Table 3 demonstrates, this intervention generally improved temporal reasoning accuracy, though its varied effectiveness across models likely reflects their limited exposure to timestamp-related data, underscoring both their inherent capabilities and future potential.

Table 2: *Emotion Recognition* Accuracy for the full dataset and the subset where target individuals are in the speaking state.

| Models | Emo. Acc. | Speaker Emo. Acc. |
|--------|-----------|-------------------|
| CogVLM2-Video | 38.4 | 34.7(↓3.7) |
| VideoLLaMA3 | 34.6 | 30.3(↓4.3) |
| LLaVAOneVision | 36.9 | 34.7(↓2.2) |
| InternVL2.5 | 37.3 | 34.1(↓3.2) |
| InternVideo | 35.7 | 31.0(↓4.7) |
| Qwen-VL2.5 | 43.0 | 41.0(↓2.0) |
| ShareGPT4Video | 35.0 | 31.6(↓3.4) |
| ChatBridge | 28.9 | 24.1(↓4.8) |
| Qwen2.5-Omni | 42.2 | 40.3(↓1.9) |
| Average | 36.9 | 33.5(↓3.4) |

Table 3: Timestamp integration effect on Video-MLLMs in *Appearance Time Detection* , *Action at Specific Time* and *Time of Specific Action*.

| Video-MLLMs | ATD | AST | TSA |
|-------------|-----|-----|-----|
| Chat-UniVi | 17.5 (↑2.3) | 40 (↑26) | 14.1 (↑7.4) |
| CogVLM2 | 40.8 (↑0.8) | 35 (↓1) | 10.7 (↓3.4) |
| VILA | 38.3 (↑18.1) | 33 (–) | 48 (↓3.4) |
| Video-LLaVA | 36.7 (↑5) | 25 (↓2) | 45.8 (↑5.1) |
| LLaVAOneVision | 10 (–) | 41 (↑4) | 47.5 (–) |
| InternVL2 | 31.7 (↑10.9) | 45 (↑7) | 52.5 (↑18.7) |
| Video-LLaMA | 24.2 (↑4.2) | 25 (↑7) | 9 (–) |
| Video-LLaMA2.1 | 24.2 (↑7.5) | 25 (↓2) | 29.4 (↓3.9) |
| ChatBridge | 17.5 (↑8.8) | 40 (↓1) | 14.1 (↓2.8) |
| Average | 26.8 (↑5.9) | 34.3 (↑4.4) | 30.1 (↑2.0) |

Figure 4: Two examples of 8-frame speaker videos sampled at equal intervals in the emotion recognition task, along with the responses from different MLLMs.

## 4.4 QUANTIFYING THE REDUCTION OF MANUAL LABOR IN QUESTION GENERATION

We report the distribution of manual work required for the 6 descriptive multiple-choice questions generated by the pipeline in Table 4. On average, 72% of the generated questions contain the correct answer, requiring no revision, with humans only needing to select the correct answer. The remaining 28% lack a correct answer and require a simple manual input, typically one sentence, which is much faster than creating questions from scratch. Notably, the overall proportion of questions containing the correct answer exceeds the best model's accuracy in the final evaluation (e.g., the "no-edit" rate for ER is 81%, while the best model accuracy is 49.4%), as options are drawn from multiple models, increasing correct answer coverage and reflecting the benefit of ensemble-based de-

Table 4: Effectiveness of the Distractor-Included QA Synthesis Pipeline in generating six types of descriptive multiple-choice questions. The Efficiency Gain is calculated as 1 / (1 - No Edit Rate).

| Question Type | No Edit | Efficiency Gain |
|---------------|---------|-----------------|
| *Behavior Temporal Analysis* | 68% | 3.1 |
| *Emotion Intensity Comparison* | 63% | 2.7 |
| *Emotion Recognition* | 81% | 5.3 |
| *Behavior Causality Analysis* | 74% | 3.8 |
| *Emotion Temporal Analysis* | 83% | 5.9 |
| *Human-to-Text* | 65% | 2.9 |
| Average | 72.3% | 3.6 |

sign. Besides, incorrect model responses are repurposed as plausible distractors, enhancing question difficulty. For other rule-based non-descriptive tasks, over 70% of the questions can be directly adopted. Overall, while human oversight is still needed, our partial automation significantly boosts efficiency, allowing a person using the pipeline to complete 3 times more questions within the same timeframe as manual creation from scratch, substantially reducing manual workload.

## 4.5 GENERALIZABILITY OF THE SYNTHESIS PIPELINES BEYOND HUMAN-CENTRIC TASKS

A key contribution of our work is the general-purpose nature of our automated pipelines, which extends far beyond the human-centric domain. Its modular architecture allows for easy adaptation to new domains by simply swapping initial detectors. We demonstrate this by generating diagnostic questions for non-human entities, such as fine-grained attribute recognition for pets and temporal tracking for vehicles, with examples shown in Figure 5.

For object attribute recognition (e.g., identifying a specific dog, Figure 5 (a)), we adapt our pipeline by: *(i)* substituting the human detector with a dog detector (YOLOv11) to extract instance tracks (*Label-1*); *(ii)* using an MLLM to annotate attributes (fur pattern, accessories) from cropped tracks; *(iii)* selecting a uniquely described dog as the ground truth; and *(iv)* rendering bounding boxes from *Label-1* as candidates. Similarly, for object temporal tracking (e.g., a car's appearance interval,

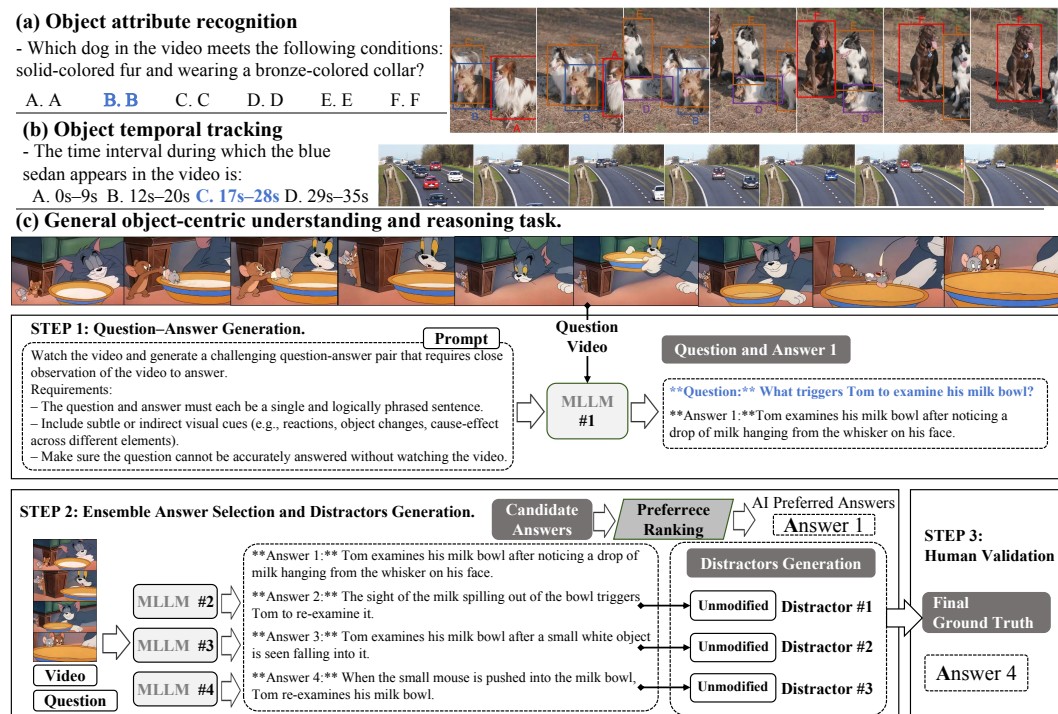

Figure 5: Demonstration of our pipeline's generalizability. (a) Object attribute recognition for pets. (b) Object temporal tracking for cars. (c) General causal reasoning QA generated for an object-centric event, with the MLLMs used in this case being Gemini-1.5-pro, VideoLLaMA3, LLaVA-Video, and gpt-4o. This highlights the adaptability of our framework.

Figure 5 (b)), we: *(i)* replace the human detector with a vehicle detector to generate tracks; *(ii)* annotate attributes (e.g., color, type)for each track; *(iii)* select a unique car as ground truth; and *(iv)* derive its appearance interval from track frames as ground truth option.. These procedures parallel our *Text-to-Human* and *Appearance Time Detection* tasks, proving the framework's broad applicability to object-centric attribute recognition and temporal tracking.

Furthermore, our Distractor-Included QA Generation Pipeline is inherently transferable, enabling the synthesis of a broad spectrum of descriptive questions and answer choices beyond human-centric scenarios, as exemplified by the general causal reasoning question in Figure 5 (c). This demonstrated transferability underscores the broader value of our methodology as a scalable solution for accelerating the creation of high-quality MLLM evaluations across diverse domains.

## 5 CONCLUSION

We present HUMANVBENCH to address the pressing need for improved assessment of human-centric video understanding in MLLMs. By incorporating extensive evaluation dimensions through 16 fine-grained tasks, HUMANVBENCH provides a systemic view into both successes and critical shortcomings of video MLLMs, particularly in emotion perception and speech-visual alignment. Experimental findings across over twenty leading video MLLMs illustrate that while proprietary models approach human accuracy in some tasks, substantial advancements are required, particularly in cross-modal domains where alignment between speech and visual elements proves challenging.

Beyond the benchmark itself, the core contribution of this work lies in its extensibility and efficient methodology. Our framework transforms model-generated errors into high-quality distractors and shifts the human role from creating tasks from scratch to efficiently validating and correcting them, thereby multiplying the benchmark's production efficiency. By open-sourcing the benchmark and underlying methodologies, we hope HUMANVBENCH can foster collaborative efforts aimed at advancing the frontiers of human-like video understanding capabilities in MLLMs.

## 6 ETHICS STATEMENT

This work adheres to the ICLR Code of Ethics. We recognize the ethical implications of human-centric video understanding with MLLMs. The HUMANVBENCH benchmark and its automated synthesis framework were designed to mitigate bias and avoid harmful stereotypes. Human oversight is incorporated in the verification stage to reduce bias propagation. No private or sensitive data were collected; all data are either publicly available or generated under controlled conditions. We release the benchmark and methodology with the intention of promoting responsible AI research, while cautioning against misuse or discriminatory applications. The authors declare no conflicts of interest.

## 7 REPRODUCIBILITY STATEMENT

To ensure the reproducibility of this work, we provide all necessary resources and detailed information. We have open-sourced the HUMANVBENCH benchmark, including data, evaluation, and synthesis codes, available at anonymous link. Comprehensive details regarding the evaluation can be found in Appendix F. Furthermore, the specifics of the annotation and the synthesis pipeline, the precise models used, and our specific prompt designs are thoroughly described in Appendix H and I. By openly releasing our benchmark and its construction methodology, we aim to enable the research community to easily reproduce our evaluation results and build new assessments based on this methodology, fostering collaborative progress in the field.

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

**Table of Contents**

## A    OVERVIEW

In the appendix, we first provide a statement on LLM usage (Appendix B). We then present additional benchmark statistics (Appendix C) and the modality ablation experiments for VideoLLaMA2 (Appendix E). Appendix F describes further evaluation details, while Appendix G introduces the definitions and examples of the 16 tasks in HUMANVBENCH. Next, Appendix H details the implementation of each operator in the proposed *Human-Centric Annotation Pipeline*, with a worked example of the annotation process. Finally, Appendix I provides construction details of all tasks and the contributions of human annotators.

## B    THE USE OF LARGE LANGUAGE MODELS

The core research idea, including the conceptualization of HUMANVBENCH and its synthesis framework, originated entirely from the authors. Multimodal Large Language Models (MLLMs) and Large Language Models (LLMs) were then integrally utilized as a methodological component to efficiently automate data annotation and challenging question construction for the benchmark, a key focus of our research. Details of MLLM utilization, including the models, prompts, and annotation pipeline, are provided in the methodology section of the main paper (Section 3) and further elaborated in Appendix H and Appendix I. Additionally, LLMs assisted with language refinement during manuscript preparation. The authors assume full responsibility for all content, including research ideas, methodologies, and findings.

## C    MORE STATISTICS OF HUMANVBENCH

HUMANVBENCH focuses on short video understanding, specifically videos with a duration of 30-second or less. It includes a total of 2475 question instances, with the specific number for each task indicated in 1. The total video duration amounts to 5.2 hours and demonstrates a variety of people, scenes, and video shooting styles, as shown in 6.

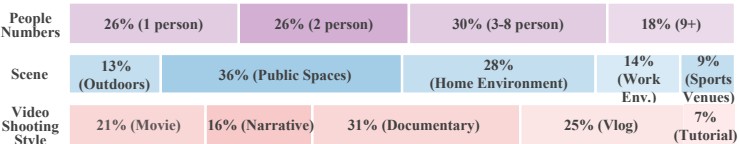

Figure 6: The distribution of the number of people, scenes, and video shooting styles in HUMAN-VBENCH

## D    FULL RESULTS FOR 24 MODELS ON 16 TASKS

We meticulously select 22 SOTA open-source video MLLMs. These included both visual-only models such as Qwen-VL series (Bai et al., 2025; Wang et al., 2024), InternVL series (Chen et al., 2024g; 2025), VideoLLaMA3, InternVideo2.5 (Wang et al., 2025a), ViLAMP (Cheng et al., 2025),LLaVA-Video (Zhang et al., 2025c), ShareGPT4Video, CogVLM2-video (Hong et al., 2024), LLaVA-One-Vision (Li et al., 2024a), Chat-UniVi (Jin et al., 2024), VILA (Lin et al., 2024b), VideoChat (Li et al., 2023), and audio-visual models capable of analyzing both visual and audio inputs, such as the Qwen2.5-Omni (Xu et al., 2025), MIO (Wang et al., 2025b), Video-LLaMA series (Zhang et al., 2023; Cheng et al., 2024) and generalist MLLMs like ImageBind-LLM (Girdhar et al., 2023), Chat-Bridge (Zhao et al., 2023), and OneLLM (Han et al., 2024).

## E    MODALITY ABLATION IN VIDEOLLAMA2

Despite audio-visual MLLMs processing audio data, they perform at random-guess levels on AVSM and ASD tasks, underperforming relative to many vision-only models that rely solely on lip movement analysis. This raises the question: does the poor performance stem from limitations in visual

Table 5: Results on HUMANVBENCH with 15 visual-only MLLMs, 7 audio-visual MLLMs, and 5 proprietary models. "Random" denotes random guessing, and "Human" indicates human-level performance. Task acronyms are defined in Figure 1. For each task, the best overall results are shown in bold, while the best open-source model results are underlined. "-" means the model recognizes its lack of required capabilities for the task and thus refuses to answer. The results for all open-source models are averaged over five runs with different random seeds.

| Models | Emotion Perception | | | | | Person Recognition | | | | | Behavior Analysis | | | | | Speech-Visual Alignment | | | | |
|---|---|---|---|---|---|---|---|---|---|---|---|---|---|---|---|---|---|---|---|---|
| | ER | ETA | AR | EIC | Avg | T2H | H2T | HC | ATD | Avg | BTA | BCA | AST | TSA | Avg | AVSM | ASD | AVAD | SCM | Avg |
| Random | 24.6 | 24.7 | 25.0 | 23.1 | 24.4 | 27.9 | 24.6 | 23.1 | 25.0 | 25.2 | 23.0 | 23.6 | 25.0 | 20.0 | 22.9 | 42.8 | 23.6 | 33.3 | 25.0 | 31.2 |
| ViLA | 32.7 | 24.8 | 28.7 | 21.4 | 26.9 | 50.4 | 27.0 | 40.4 | 20.2 | 34.5 | 33.0 | 48.6 | 33.0 | 51.4 | 41.5 | 47.4 | 23.9 | 34.3 | 18.6 | 31.1 |
| Video-LLaVA | 18.3 | 18.8 | 26.2 | 8.0 | 17.8 | 27.9 | 40.6 | 28.3 | 31.7 | 32.1 | 30.8 | 32.5 | 27.0 | 40.7 | 32.8 | 50.0 | 28.4 | 34.3 | 21.2 | 33.5 |
| Chat-Univi | 26.4 | 15.0 | 10.9 | 16.5 | 17.2 | 29.1 | 27.0 | 17.8 | 19.8 | 23.4 | 27.9 | 39.4 | 14.0 | 6.7 | 22.0 | 42.8 | 20.6 | 26.3 | 18.6 | 27.1 |
| VideoChat2-IT | 33.5 | 29.1 | 39.6 | 26.8 | 32.3 | 20.1 | 47.3 | 11.2 | 26.7 | 26.3 | 45.8 | 43.7 | 34.0 | 24.9 | 37.1 | 43.4 | 27.7 | 31.4 | 23.0 | 31.4 |
| InternVL2 | 36.1 | 31.3 | 40.2 | 30.4 | 34.5 | 70.4 | 61.2 | 37.2 | 20.8 | 47.4 | 49.8 | 52.3 | 38.0 | 33.8 | 43.5 | 51.7 | 55.0 | 33.6 | 25.5 | 41.5 |
| Qwen-VL2 | 38.0 | 35.1 | 42.7 | 37.5 | 38.3 | 79.3 | 72.7 | 43.4 | 20.8 | 54.1 | 47.8 | 55.6 | 32.0 | 51.4 | 46.7 | 50.7 | 56.1 | 31.4 | 23.7 | 40.5 |
| CogVLM2-Video | 38.4 | 29.9 | 36.0 | 25.0 | 32.3 | 42.5 | 51.5 | 31.1 | 40.0 | 41.3 | 41.9 | 50.3 | 34.0 | 14.1 | 35.1 | 59.2 | 38.7 | 30.7 | 17.9 | 36.6 |
| VideoLLaMA3 | 34.6 | 29.9 | 48.0 | 46.4 | 39.7 | 90.3 | 67.9 | 44.9 | 71.0 | 68.5 | 41.5 | 56.3 | 56.4 | 69.0 | 55.8 | 64.6 | 63.1 | 34.4 | 17.9 | 45.0 |
| LLaVAOneVision | 36.9 | 31.3 | 62.8 | 28.6 | 39.9 | 67.0 | 61.2 | 49.1 | 10.0 | 46.8 | 45.5 | 55.6 | 37.0 | 47.5 | 46.4 | 52.6 | 51.6 | 35.0 | 26.9 | 41.5 |
| InternVL2.5 | 37.3 | 38.1 | 54.3 | 37.5 | 41.8 | 81.0 | 73.3 | 40.6 | 35.8 | 57.7 | 53.0 | 57.6 | 41.0 | 62.1 | 53.4 | 65.1 | 61.3 | 32.1 | 15.4 | 43.5 |
| InternVideo | 35.7 | 42.9 | 47.6 | 35.7 | 40.5 | 64.2 | 69.1 | 50.9 | 42.5 | 56.7 | 56.1 | 58.3 | 42.0 | 55.9 | 53.1 | 59.4 | 59.4 | 32.8 | 31.6 | 41.5 |
| Qwen-VL2.5 | 43.0 | 30.6 | 35.4 | 50.0 | 39.8 | 88.8 | 74.5 | 50.9 | 30.8 | 61.3 | 51.0 | 58.3 | 34.0 | 50.8 | 48.5 | 71.1 | 61.3 | 33.6 | 18.8 | 46.2 |
| ShareGPT4Video | 35.0 | 39.1 | 40.9 | 29.5 | 36.1 | 33.0 | 38.8 | 24.5 | 43.3 | 34.9 | 32.8 | 43.7 | 39.0 | 16.9 | 33.1 | 44.1 | 31.0 | 34.3 | 25.0 | 33.6 |
| LLaVA-Video | 35.4 | 32.3 | 59.8 | 28.6 | 39.0 | 74.3 | 61.8 | 44.3 | 26.7 | 51.8 | 41.9 | 57.0 | 47.0 | 58.8 | 51.2 | 52.0 | 58.7 | 32.8 | 39.3 | 45.7 |
| ViLAMP | 41.4 | 29.3 | 48.8 | 31.2 | 37.7 | 59.8 | 62.4 | 45.3 | 20.0 | 46.9 | 55.7 | 49.7 | 52.0 | 36.7 | 48.5 | 53.3 | 51.0 | 32.3 | 31.6 | 42.1 |
| VideoLLaMA | 30.8 | 21.6 | 11.5 | 20.5 | 21.1 | 28.4 | 23.0 | 23.9 | 20.0 | 23.8 | 22.9 | 41.1 | 23.2 | 15.4 | 25.7 | 40.1 | 26.6 | 33.1 | 26.2 | 31.5 |
| VideoLLaMA2.1 | 38.0 | 30.6 | 47.0 | 30.4 | 36.5 | 34.6 | 52.7 | 41.5 | 16.7 | 36.4 | 53.8 | 60.3 | 27.0 | 33.3 | 43.6 | 44.0 | 31.6 | 32.1 | 23.7 | 32.9 |
| MIO | 29.3 | 20.3 | 29.3 | 10.7 | 22.4 | 4.5 | 18.2 | 10.4 | 23.3 | 14.1 | 22.9 | 39.7 | 29.0 | 19.2 | 27.7 | 42.1 | 31.6 | 16.1 | 19.6 | 27.4 |
| ImageBind-LLM | 21.3 | 22.4 | 25.0 | 11.6 | 20.1 | 23.5 | 13.3 | 23.8 | 19.5 | 20.0 | 15.4 | 32.4 | 24.0 | 23.3 | 23.8 | 45.0 | 25.1 | 28.9 | 22.9 | 30.5 |
| ChatBridge | 28.9 | 18.0 | 30.2 | 16.1 | 23.3 | 31.4 | 30.9 | 23.7 | 8.7 | 23.7 | 24.1 | 43.7 | 41.0 | 16.9 | 31.4 | 43.7 | 25.3 | 30.4 | 24.4 | 31.0 |
| OneLLM | 27.0 | 26.1 | 34.1 | 7.1 | 23.6 | 29.0 | 36.4 | 20.6 | 27.8 | 28.5 | 24.9 | 49.0 | 23.0 | 22.4 | 29.8 | 43.4 | 26.4 | 29.5 | 23.2 | 30.6 |
| Qwen2.5-Omni | 42.2 | 30.1 | 36.6 | 33.0 | 35.5 | 56.4 | 64.2 | 41.5 | 15.8 | 44.5 | 45.8 | 55.0 | 28.0 | 24.3 | 38.3 | 71.1 | 48.4 | 27.0 | 71.8 | 54.6 |
| GPT-4o (20241120) | 36.5 | 43.6 | 25.6 | 28.5 | 33.6 | 49.2 | 77.0 | 47.2 | 32.5 | 50.9 | 62.1 | 62.9 | 52.0 | 71.2 | 62.1 | - | - | - | - | - |
| GPT-5 (20250807) | 43.7 | 48.9 | 59.8 | 34.8 | 46.8 | 69.8 | 84.8 | 64.2 | 58.3 | 69.5 | 73.1 | 66.9 | 64.0 | 65.0 | 67.3 | - | - | - | - | - |
| Gemini-1.5-Pro | 49.4 | 51.9 | 53.0 | 49.1 | 50.9 | 87.1 | 73.9 | 52.8 | 71.7 | 71.4 | 53.4 | 60.3 | 54.0 | 75.1 | 60.7 | 90.1 | 76.8 | 66.4 | 84.6 | 79.5 |
| Gemini-2.5-flash | 52.1 | 57.1 | 59.1 | 43.8 | 53.0 | 92.2 | 77.6 | 61.3 | 71.7 | 75.7 | 65.6 | 68.9 | 58.0 | 67.2 | 64.9 | 98.7 | 80.6 | 57.7 | 99.1 | 84.0 |
| Gemini-2.5-Pro | 54.4 | 54.1 | 53.0 | 50.0 | 52.9 | 95.0 | 84.8 | 69.8 | 84.2 | 83.5 | 79.0 | 72.2 | 63.0 | 68.4 | 70.7 | 96.7 | 78.7 | 72.3 | 98.3 | 86.5 |
| Human | 87.6 | 85.0 | 87.8 | 78.0 | 84.6 | 98.9 | 84.1 | 92.5 | 78.3 | 88.5 | 86.7 | 84.5 | 88.6 | 88.1 | 87.0 | 96.0 | 96.1 | 87.0 | 98.3 | 94.4 |

Table 6: The performance of VideoLLaMA2's vision-only version (VideoLLaMA2-7B-16F), and the audio-visual version (VideoLLaMA2.1-AV), on HUMANVBENCH, based on different modal inputs (A for Audio, V for Visual).

| Models | Input Modal | Speech-Visual Alignment | | | | |
|---|---|---|---|---|---|---|
| | | AVSM | ASD | AVAD | SCM | Avg |
| Random | | 42.8 | 23.6 | 33.3 | 25.0 | 31.2 |
| Video-LLaMA2.1-AV | A, V | 44.0 | 31.6 | 32.1 | 23.7 | 32.9 |
| Video-LLaMA2.1-AV | V | 43.4 | 29.0 | 32.8 | 18.8 | 31.0 |
| VideoLLaMA2-7B-16F | V | 47.4 | 38.1 | 32.1 | 15.4 | 33.3 |

analysis (e.g., lacking lip-reading ability) or from the interference of audio input? To explore this, we conducted ablation experiments using the VideoLLaMA2 model series, chosen for its open-source availability of both vision-only and audio-visual variants.

As shown in the Table 6, VideoLLaMA2-7B-16F (vision-only) exhibits only a slight advantage over Video-LLaMA2.1-AV (audio-visual) on AVSM and ASD tasks, yet still lags far behind vision-only models such as VideoLLaMA3 and InternVL2.5 (Table 5). This indicates that VideoLLaMA2-7B has inherently poor lip-reading capability, which further implies that the audio-visual variant (Video-LLaMA2.1-AV) also suffers from limited visual lip-reading ability. Such limitations constrain its upper-bound performance in speech-visual alignment tasks. On the other hand, Video-LLaMA2.1-AV shows no significant performance advantage when utilizing audio information compared to its vision-only counterpart. This suggests that vocal information is not effectively leveraged, likely due to insufficient speech parsing capability in video MLLMs and inadequate understanding of cross-modal associations between audio and visual content.

## F  MODEL EVALUATION IMPLEMENTATION

**Prompt.** In order to facilitate the statistical model to answer the results, following common practices used in MLLM evaluations (Jiao et al., 2025b), we adopt the following prompt to guide the MLLM to

output option letters: "*Select the best answer to the following multiple-choice question based on the video. Respond with only the letter of the correct option. <Question-choices>   Only answer best answer's option letter. Your option is:*   ". **Evaluation Environments.** All evaluation experiments for open-source models were conducted on a single NVIDIA L20 GPU with an inference batch size of 1.

**Baseline Configurations and Runtime Statistics.** Table 7 shows the scale, parameter settings, and costs (including memory usage and end-to-end testing time) for each model on HUMANVBENCH. All hyperparameter settings follow the default configurations of these open-source works.

| Model | Time (min) | top_p | top_k | num_beams | temp. | frames |
|---|---|---|---|---|---|---|
| Chat-UniVi (7B) | 40 | 1 | 50 | 1 | 0.2 | 1 f/s |
| CogVLM2-Video (8B) | 37 | 0.1 | 1 | 1 | 0.2 | 1 f/s |
| Video-LLaVA (7B) | 49 | 1 | 50 | 1 | 1 | 8 f |
| LLaVA-OneVision (7B) | 49 | 1 | 50 | 1 | 1 | 8 f |
| PLLaVA (7B) | 41 | 0.9 | 50 | 1 | 1 | 16 f |
| ShareGPT4Video (8B) | 50 | 0.9 | 50 | 1 | 1 | 16 f |
| Otter-V (7B) | 62 | 1 | 50 | 3 | 1 | 16 f |
| VideoChat2-IT (7B) | 53 | 0.9 | 50 | 1 | 1 | 16 f |
| InternVL2 (7B) | 44 | 1 | 52 | 1 | 1.0 | 8 f |
| InternVL2.5 (7B) | 31 | 1 | 52 | 1 | 1.0 | 8 f |
| Qwen2-VL (7B) | 45 | 0.001 | 1 | 1 | 0.1 | 2 f/s |
| Qwen2.5-VL (7B) | 35 | 0.001 | 1 | 1 | 0.1 | 2 f/s |
| LLaVA-Video (7B) | 182 | 0.8 | 20 | 1 | 0.7 | 64 f |
| Video-LLaMA3 (7B) | 90 | 0.8 | 20 | 1 | 0.7 | 1 f/s |
| InternVideo2.5 (7B) | 170 | 1 | 50 | 1 | 1 | 128f |
| ViLAMP (7B) | 29 | 0.8 | 20 | 1 | 0.7 | 1 f/s |
| Video-LLaMA (7B) | 40 | 1 | 50 | 2 | 1 | 8 f |
| Video-LLaMA2.1 (7B) | 31 | 0.9 | 50 | 1 | 0.2 | 8 f |
| ImageBind-LLM (7B) | 77 | 1 | 50 | 1 | 1 | 15 f |
| ChatBridge (13B) | 29 | 1 | 50 | 1 | 0.2 | 4 f |
| OneLLM (7B) | 130 | 0.75 | 50 | 1 | 0.1 | 15 f |
| MIO (7B) | 88 | 0.7 | 0 | 1 | 1 | FPS/10 |
| Qwen2.5_Omni (7B) | 91 | 1 | 50 | 1 | 1 | 2 f/s |
| GPT-4o (20241120) | 191 | 1 | - | - | 1 | FPS/10 |
| Gemini-1.5-Pro | 254 | 1 | 40 | - | 0.9 | API |

Table 7: Model configuration and runtime statistics evaluated on HUMANVBENCH (one pass for all provided test samples).

# G   DEFINITIONS AND EXAMPLES FOR EACH TASK

## G.1   EMOTION PERCEPTION

**Emotion Recognition** aims to judge the overall emotional state of the person highlighted by a red bounding box in the video. An example is shown in Figure 7.

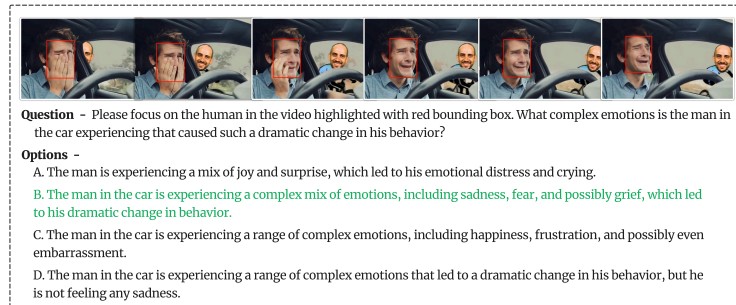

Figure 7: Example of Emotion Recognition task.

**Emotion Temporal Analysis** involves analyzing the changes in the emotions of the people highlighted with the red bounding box over time, identifying gradual intensification, diminishment, emotions shifts to test the model's ability to track emotional dynamics. An example is shown in Figure 8.

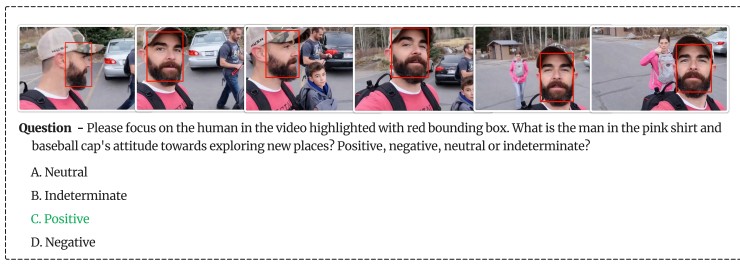

**Question** – Please focus on the human in the video highlighted with red bounding box. How does the man's emotional state evolve throughout the video? Please describe the sequence of emotions.

**Options** –

A. The man starts off feeling curious and surprised, then becomes happy and excited at certain points, and finally to a state of relief.

B. The man starts off feeling curious and surprised, then becomes happy and excited at certain points, and finally to a state of relief.

C. The man's emotional state evolves from  neutral to angry.

D. The man's emotional state evolves from neutral to surprised, then to shocked, and finally to a mix of shock and confusion.

Figure 8: Example of Emotion Temporal Analysis task.

**Attitude Recognition** involves inferring a character's attitude towards things, categorized into four fixed options: positive, neutral, negative, and indeterminate. An example is shown in Figure 9.

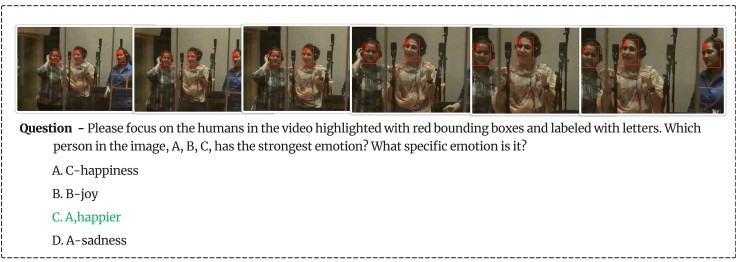

**Question** – Please focus on the human in the video highlighted with red bounding box. What is the man in the pink shirt and baseball cap's attitude towards exploring new places? Positive, negative, neutral or indeterminate?

A. Neutral

B. Indeterminate

C. Positive

D. Negative

Figure 9: Example of Attitude Recognition task.

**Emotion Intensity Comparison** requires compares the emotional intensity differences among various individuals in the video to find the most emotional person, assess whether the model can quantify and differentiate emotional intensity. An example is shown in Figure 10.

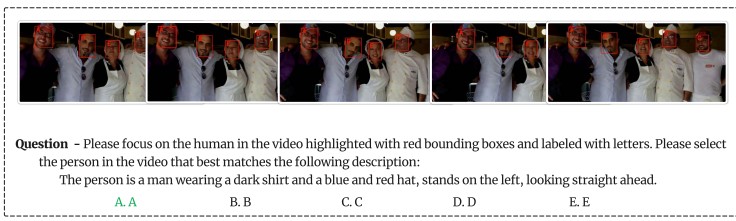

**Question** – Please focus on the humans in the video highlighted with red bounding boxes and labeled with letters. Which person in the image, A, B, C, has the strongest emotion? What specific emotion is it?

A. C–happiness

B. B–joy

C. A,happier

D. A–sadness

Figure 10: Example of Emotion Intensity Comparision task.

## G.2 PERSON RECOGNITION

**Text-to-Human** requires the model to identify the specific person in a multi-person video based on a given text description, to test the model's ability to locate and identify the described person. An example is shown in Figure 11.

**Question** – Please focus on the human in the video highlighted with red bounding boxes and labeled with letters. Please select the person in the video that best matches the following description:

The person is a man wearing a dark shirt and a blue and red hat, stands on the left, looking straight ahead.

A. A          B. B          C. C          D. D          E. E

Figure 11: Example of Text-to-Human task.

**Human-to-Text** asks the model to choose the most accurate description of the target person in a multi-person video, to ensure that the person is clearly distinguished from others and uniquely identified. This task requires the model to analyze and compare individuals in the video, identifying distinguishing features of the target person, such as appearance, clothing, actions, location, and other characteristics. An example is shown in Figure 12.

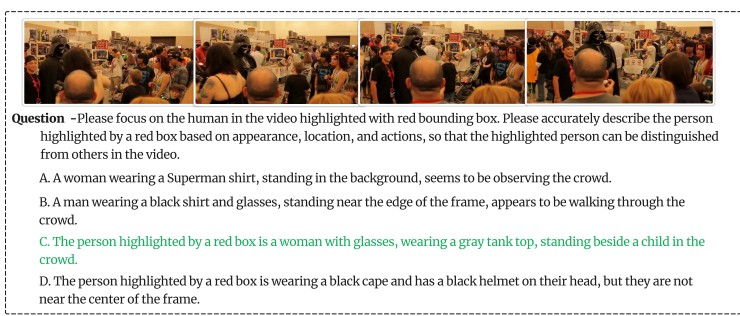

Figure 12: Example of Human-to-Text task.

**Human Counting** requires the model to determine the total number of distinct individuals in the video, testing its capability to detect, track, and accurately count individuals in complex scenes. An example is shown in Figure 13.

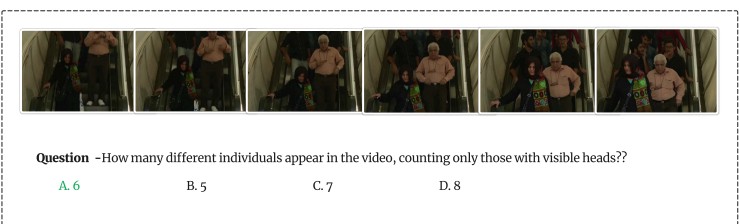

Figure 13: Example of Human Counting task.

**Appearance Time Detection** requires the model to identify the exact time frames when a specified person appears, demanding the ability to precisely mark the start time, end time, and duration of the individual's presence in the video. An example is shown in Figure 14.

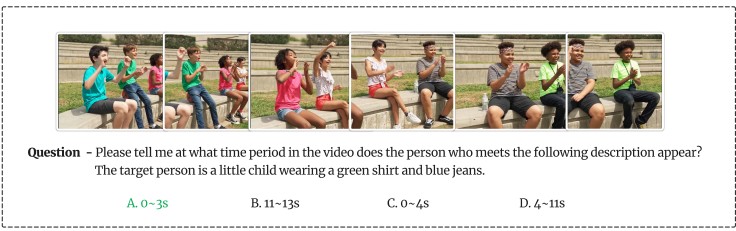

Figure 14: Example of Appearance Time Detection task.

## G.3 HUMAN BEHAVIOR ANALYSIS

**Behavior Temporal Analysis** involves analyzing the dynamic changes in a specified person's behavior over time, testing the model's ability to accurately capture and track the temporal characteristics of these changes. An example is shown in Figure 15.

**Behavior Causality Analysis** aims to investigate the causal relationships underlying a specific behavior, requiring the model to determine whether a person's behavior in the video is triggered by a particular event or leads to subsequent actions. An example is shown in Figure 16.

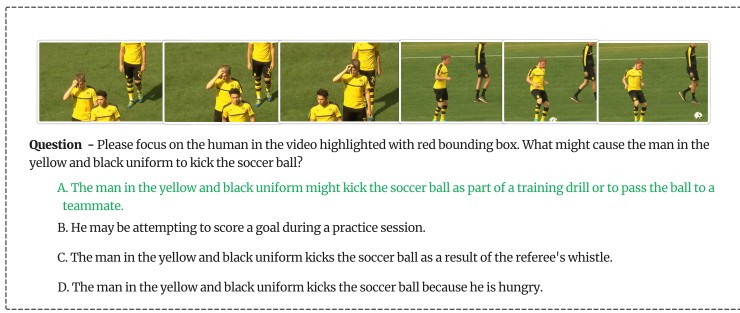

Figure 15: Behavoir Temporal Analysis task.

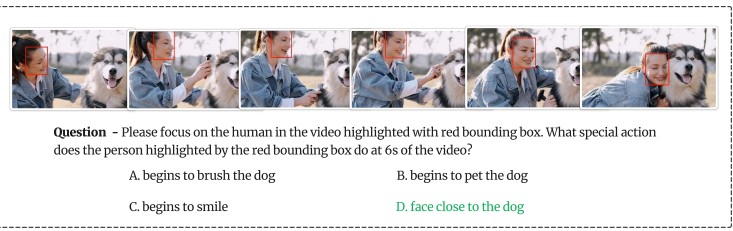

Figure 16: Example of Behavior Causualty Analysis task.

**Action at Specified Time** asks the model to identify a person's behavior or state at a specific time, testing its ability to accurately determine the person's action or state at the given moment. An example is shown in Figure 17.

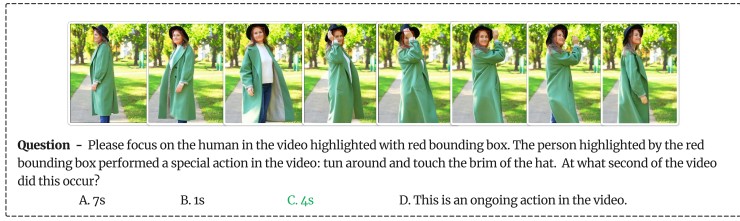

Figure 17: Example of Action at Specific Time task.

**Time of Specific Action** focuses on determining the time when a specific behavior occurs, requiring the model to accurately pinpoint the time of a particular action in the video. An example is shown in Figure 18.

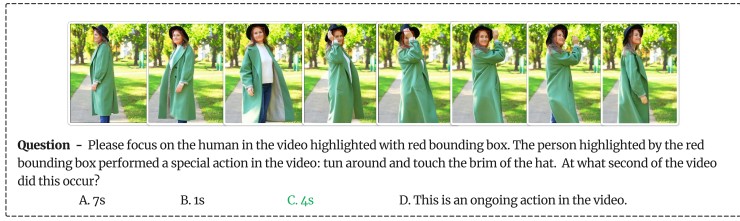

Figure 18: Example of Time of Specific Action task.

### G.4    CROSS-MODAL SPEECH-VISUAL ALIGNMENT

involves analyzing audio cues in multi-person videos to identify the individual whose appearance matches the voice. This task evaluates whether the model can recognize the voice gender and age and compare them with the appearance of the person in the video. An example is shown in Figure

19.

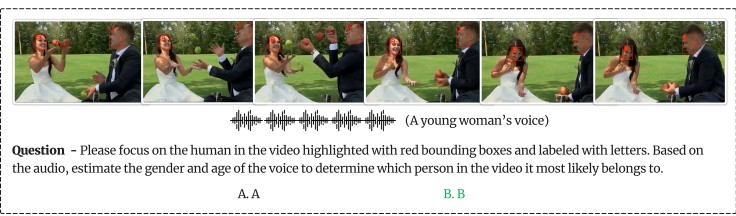

Figure 19: Example of Audio-Visual Speaker Matching task.

**Active Speaker Detection** asks the model to identify the active speaker in the video, requiring the model to accurately identify who is speaking by combining audio cues with the characters' lip movements. An example is shown in Figure 20.

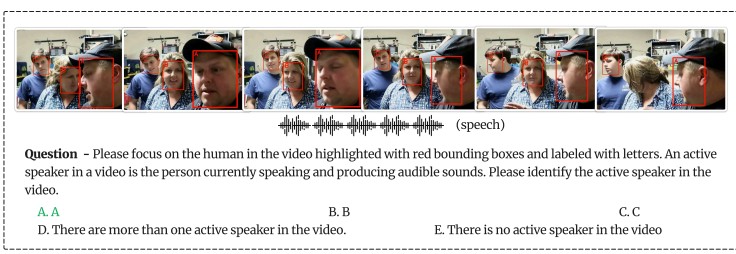

Figure 20: Example of Active Speaker Detection task.

**Audio-Visual Alignment Detection** requires detecting when the audio and video are synchronized, evaluating the model's ability to synchronize audio and visual content, particularly through analyzing the speaker's lip movements and voice. An example is shown in Figure 21.

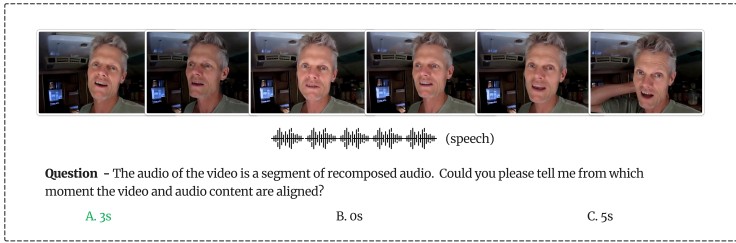

Figure 21: Example of Audio-Visual Alignment Detection task.

**Speech Content Matching** requires matching the speech content of the video with text, validating the model's ability to transcribe speech or translate lip movements into text. Figure 22 shows an example.

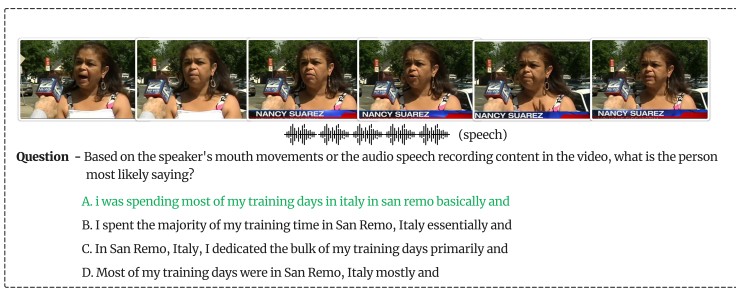

Figure 22: Example of Speech Content Matching task.

## H   ANNOTATIONS DETAILS AND EXAMPLES IN *Human-Centric Annotation Pipeline*

For the in-the-wild videos collected from Pexels, we first apply splitting and filtering operations. Specifically, we begin by utilizing the `video_resolution_filter`, `video_aesthetics_filter`, and `video_nsfw_filter` operators to select videos that meet the following criteria: a resolution of at least 1280 in width and 480 in height, acceptable aesthetics, and appropriate content. Next, the `video_split_by_scene_mapper` is used to split the videos into scenes. The resulting clips are then filtered using the `video_duration_filter` to exclude clips shorter than 1 second and the `video_motion_score_filter` to remove static videos. These steps utilize existing operators in Data-Juicer (Chen et al., 2024b), with parameters set to their default values except for the `video_motion_score_filter`, where the minimum motion score is set to 1.2. After completing these foundational steps, we apply the `video_face_ratio_filter` with a threshold of 0.65 to retain videos containing people. These videos are then processed using a series of mappers to generate fine-grained, multi-modal, human-related annotations.

We use a video example to demonstrate the annotation process and results, as shown in Figure 23. Below, we detail the models and settings used for each operator.

For the `video_human_tracks_extraction_mapper`, we follow the approach of Light_ASD (Liao et al., 2023), utilizing S3FD (Zhang et al., 2017) as the face detector. A face bounding box is added to a human track if its overlap rate exceeds 50%. After obtaining the face track, we identify the corresponding body bounding box for each face bounding box in the same frame to generate a second bounding box track for the individual, referred to as the body track. The matching criterion selects the candidate bounding box with the smallest horizontal center distance and a smaller area. This process can be expressed by the following formula:

$$
\text{closest\_bbox} = \underset{\text{bbox} \in \text{candidate\_bboxes}}{\arg\min}
$$
$$
\left( \left( \frac{x_1 + x_2}{2} - \frac{\text{f\_x1} + \text{f\_x2}}{2} \right)^2 + (x_2 - x_1)(y_2 - y_1) \right), \tag{1}
$$

where f_x1,f_x2 are the left and right boundaries of the face bounding box, the candidate human bounding boxes are obtained using YOLOv8-human, $x_1$, $x_2$, $y_1$, $y_2$ are the boundary values of a candidate human bounding box. If no bounding box meets the criteria, the frame is skipped, and detection proceeds with the other frames. Finally, the empty elements in the body track are replaced with the average of the bounding boxes from the surrounding frames.

In the `human_demographics_mapper`, we use DeepFace to perform frame-level detection of facial gender, age, and race. The analysis is conducted on cropped frames obtained directly from the `video_human_tracks_extraction_mapper` results. Finally, for a given face track, the demographics features are determined by taking the mode of the frame-level gender and ethnicity detections, and the median of the age detections.

In the `video_human_description_mapper`, we use the body bounding box track to crop the video, creating a reconstructed video focused on a single individual. This reconstructed video is then processed using ShareGPT4Video (Chen et al., 2024e) for appearance description and simple actions.

In the `video_facial_description_mapper`, we use the face bounding box track to crop the video, creating face-focused reconstructed videos for emotion description using Qwen2.5-VL (Cheng et al., 2024).

The `audio_tagging_mapper` is a built-in operator in Data-Juicer, which we use directly for audio type classification.

The core model for the operator `active_speaker_detection_mapper` is ASD-Light (Liao et al., 2023). Each face track sequence is analyzed together with the corresponding audio segment for the same time period. The model outputs a score sequence of the same length as the face track's frames, where each score evaluates whether the individual is speaking in the current frame. Positive scores indicate active speaking, while negative scores indicate not. To assign a binary "speak or not" label to a human track, we classify an individual as an active speaker if the longest sequence

of consecutive positive scores exceeds 12 frames. Notably, to reduce false positives, we cross-check the voice-based gender and age attributes with the individual's demographic features. If there is a significant mismatch, the positive label is reassigned as negative.

The automatic speech recognition model used in the `ASR_mapper` is SenseVoice (An et al., 2024), which can also be utilized in the `speech_emotion_recognition_mapper`. For the `voice_demographics_mapper`, we use the wav2vec2 (Baevski et al., 2020) model.

The results of the `video_description_mapper` are not directly involved in the construction of multiple-choice questions in this work. However, the environment, atmosphere, and events occurring in the video play a crucial role in understanding the actions and expressions of individuals. Therefore, we have included this mapper in the Human-Centric Annotation Pipeline. The example shown in Figure 23 is generated by ShareGPT4Video.

Notably, in the Human-Centric Video Annotation Pipeline, all the models we use are based on the most advanced open-source models available. As more powerful and specialized models emerge, integrating them into our pipeline can further enhance the quality of annotations.

# I   COMPLETE CONSTRUCTION DETAILS OF ALL TASKS

We will first explain the details of six descriptive questions generated using the Distractor-Included QA Generation Pipeline, followed by the construction details of the remaining tasks.

## I.1   CONSTRUCTION DETAILS OF 6 DESCRIPTIVE HUMAN-CENTRIC QUESTIONS

We first present the general instruction templates in the six task generation processes.

The prompt template for the three Video-MLLMs used to refine answers and generate raw distractors is:
*Please focus on the people whose heads are highlighted with red bounding boxes in the video and answer my question: ⟨Question⟩; Provide a brief response in one sentence.*

The instruction used to compare the answers is:
*Based on the video and my question: ⟨Question⟩Tell me which answer is better: (A). ⟨current model's answer. ⟩(B). ⟨previous best answer⟩. Just answer (A) or (B).*

The prompt template used in the "LLM for Generating Distractors" in Figure 3 is:
*Below is a ready-made question and its multiple-choice options: ⟨Question⟩, Proper Answer: ⟨Answer⟩, Distractors1: ⟨eliminator1⟩, Distractors2: ⟨eliminator2⟩, Distractors3: ⟨eliminator3⟩. This question-option set may have the following issue: The current distractors have no errors; they simply represent alternative answers to the question. This makes the correct answer less distinct compared to the distractors. Therefore, I would like your help to add minor, distinct errors to each distractor so that the correct answer is clearly the only Proper Answer. Here are the minor errors type available for selection: ⟨error type⟩.*
*Remember that the modified distractors must meet the following requirements: 1. Be modified from the original distractor with only slight changes. You are not allow to creat new ones from scratch. 2. Be distinctly different from the Answer, without being overly semantically similar. Minor errors can be added. 3. Differ from each other. 4. Distractors should have similar length to the correct answer. If it is too short, lengthen the description.*

In addition to the questions and options, the differences include the ⟨error types⟩. Next, we describe the construction of each task in detail.

**Emotion Recognition**: Since *Label-5* is naturally a description based on face-focused cropping videos, it is directly used as the task-oriented caption. Additionally, *Label-4* is included in the task-oriented caption to enhance the detail of the questions. Considering that most in-the-wild videos exhibit positive or neutral emotions, while we aim to ensure a sufficient proportion of negative emotions in the evaluation, videos for question generation are preselected at a ratio of positive:neutral:negative = 1:1:2. This selection is achieved by using an LLM to classify the emotional polarity of the descriptions. The resulting balanced category captions are used for question generation, with the following prompt:

*Please generate one question and answer pair based on the person's description: ⟨task-oriented caption⟩. the question should closely related to emotion recognition. Here is an question example: "What emotions might the girl in red dress be experiencing during her practice?"*

The video for the question is marked using the face bounding box from the target character's *Label-1*. The type of minor errors (⟨error types⟩) introduced for building distractors is: *Add incorrect emotional descriptors or modify the original emotional descriptors to incorrect ones.*

**Emotion Temporal Analysis**: We first select videos longer than 7 seconds and then identify described characters with emotional changes based on *Label-5* (using an LLM for binary classification). The videos of these characters are used for question generation. For this task, *Label-5* is directly used as the task-oriented caption, with *Label-4* added to enhance the details of the questions. The question generation prompt is:

*Please generate a question-answer pair based on the following video caption. Please note that the questions must be related to the emotional temporal changes. Here are some example: 1. How does the girl in red's emotions change as the video progresses? 2. How does the girl in red's emotions change as she dances in the video?*

The video for the question is marked using the face bounding box from the target character's *Label-1*. The type of minor errors introduced for building distractors is: *Add some incorrect emotions to the sequence, remove some correct emotion words, or change the original emotional descriptors to incorrect ones.*

**Behavior Temporal Analysis**: First, videos longer than 7 seconds are selected for question generation. Then, the target character in the video is highlighted using the face bounding box track from *Label-1*. Based on the marked videos, appearance cues of the target character (i.e., *Label-4*) are added to help to guide the model's attention to the individual. The prompt for obtaining the task-oriented caption is designed as follows:

*Please focus on the person highlighted by the red bounding box (⟨Human_Appearance⟩) and tell me if the actions of the person changed over time and what actions does the person take in order? Respond according to the following format: {"Action_Change": True or False, "Action_Sequence": action sequence}.*

Based on the task-oriented captions, select characters with changes in actions for LLM question generation. The prompt for question generation is: *Please generate a question-answer pair based on the following human's behavior caption. The generated questions should focus on identifying the action sequence of the highlighted person. The following is a description of a human in the video. ⟨action_sequence⟩. The focused person is ⟨appearance⟩. Here is a question template you can refer to: What actions and behaviors does the girl in the red dress display in the video in order? List them sequentially. Remember do not reveal the answers in your questions and the answer should be brief and just in one sentence.*

The type of minor errors introduced for building distractors is: *Add some nonexistent actions, remove some actions, or replace correct actions with incorrect ones.*

**Emotion Intensity Compare**: First, count the number of frames corresponding to the track with the longest appearance time in each video. If there are more than three and less than seven tracks in the video that reach this number of frames, keep the video for question generation. This step mainly use the information from *Label-1* . All individuals corresponding to the tracks with the most frames will be used for question generation. The question video is created by utilizing these human tracks to mark the individuals and adding letter labels. For this task, initial question-answer pairs are directly created. The question is, "Which person in the image, ⟨LETTERS⟩, has the strongest emotion? What specific emotion is it? Please respond briefly in the format ⟨letter-emotion⟩.", in which ⟨LETTERS⟩ refers to all the selectable individuals' letter labels. The answer is, "The emotional intensity of the selectable characters in the image is similar, and they are all neutral." The subsequent three models will refine the answer.

The type of minor errors introduced for building distractors is: *If the letters are the same, minor modifications to the emotions can be made to make the options different; if the letters referring to people are different, the emotions can remain unchanged.*

**Human-to-Text**: First, select videos with 3 to 7 individuals based on *Label-2*, and then choose the person who appears the most frames in the video as the target individual for question generation. Next, highlight the target individual in the video using the face bounding box track from *Label-1*.

Based on the marked video, appearance cues of the target individual (i.e., *Label-4*) are added to help the model focus on the person. The prompt for obtaining the task-oriented caption is designed as follows:

*Please accurately describe the person highlighted by a red box(⟨appearance⟩), your answer can be based on appearance, location, and actions, so that the highlighted person can be distinguished from others in the video. Please respond in only one sentence and begin with "The person is ...".*

Based on the above description of the target individual, the question-answer pair is directly constructed. The question is fixed as: "Please accurately describe the person highlighted by a red box based on appearance, location, and actions, so that the highlighted person can be distinguished from others in the video." The initial answer is the task-oriented caption.

The type of minor errors introduced for building distractors is:*Based on the items, people, and position information, add small modifications to make the location information incorrect; alternatively, you can also modify the description of the person's appearance to introduce errors.*

**Behavioral Causality Analysis**: The construction process is similar to the design process of Behavior Temporal Analysis. First, videos longer than 7 seconds are selected for question generation. Then, the target individual in the video is highlighted using the face bounding box track from *Label-1*. Based on the annotated video, appearance cues of the target individual (i.e., *Label-4*) are added to assist the model in identifying the person to focus on. The prompt for obtaining the task-oriented caption is designed as follows:

*Please describe the causal events related to the person highlighted by the red bounding box (⟨appearance⟩) in the video: what causes this person to exhibit a certain behavior, or what actions does this person take that led to a certain event. If no causal events exist, respond without causal events. Please answer in the following format: {"causal_events_exist": True or False, "causal_events_description": description}.*

Videos and target individuals with causal relationships (i.e. "causal_events_exist" is true) are then selected for question generation. The prompt for question generation is: *Please generate a question-answer pair based on the following video caption. The generated questions should inquire about causal reasoning related to the character's expressions or behaviors. The following is a description of the human in the video: ⟨causal_events_description⟩. The focused person is ⟨appearance⟩. You should either follow the causal analysis question template "Analyze why the girl in the red dress raises her hand." or the result derivation question template "What does the girl in the red dress raising her hand lead to?". Remember do not reveal the answers in your questions and the answer should be brief and just in one sentence.*

The type of minor errors introduced for building distractors is: *Explain the result using incorrect causes, misdescribe the effect of the cause-and-effect relationship, reverse the order of cause and effect, exaggerate or minimize factors.*

## I.2   CONSTRUCTION DETAILS OF 10 CLOSED-ENDED HUMAN-CENTRIC QUESTIONS

**Attitude Recognition**: This task is constructed based on the first half of the Distractor-Included QA Synthesis Pipeline. The human who appears in the most frames is selected as the target for question generation. The target individual is highlighted in the video using the face bounding box track from *Label-1*. Based on the annotated video, appearance cues of the target individual (i.e., *Label-4*) are added to the prompt to help the model focus on the intended person. The prompt used to obtain the task-specific caption is:

*Focus on the person highlighted by the red bounding box (⟨appearance⟩) and tell me: Do the highlighted people display certain attitudes toward specific objects and events? What kind of attitude is it?*

The prompt used for question generation is:

*Please generate a best question-answer pair based on the following video caption. The generated questions should focus on analyzing the character's attitude, which should be one of positive, negative, or neutral. The following is a description of the human in the video. ⟨task-specific caption⟩The focused person is ⟨appearance⟩. Here are some question templates you can refer to: 1. What is the attitude of the girl in the blue shirt towards taking the bus in the video? positive, negative, or neutral? 2. What is the woman in the beige jacket's attitude? Positive, negative, neutral? Please remember not to reveal the answers in your questions and the answer should be brief and just in one sentence.*

The options consist of four choices: Positive, Negative, Neutral, and Indetermi-

nate option is included as a supplemental choice to ensure answers optional.

**Text-to-Human**: The criteria for selecting the videos for questioning are consistent with the Emotion Intensity Compare task selection rules. Then, use the same method as Human-to-Text to obtain task-specific captions and directly use the description of the target person to complete the question template: "Please select the person in the video that best matches the following description: ⟨human description ⟩". The video corresponding to the question is marked with the face bounding boxes of all individuals using *Label-1*, and each individual is distinguished by a capital letter label. The selectable options are the letter labels representing each person.

**Human Counting**: For an annotated video, the approximate number of people in the video can be estimated directly using *Label-2*. However, due to issues such as blurred crowd background, overlapping between people and objects and other factors, this estimate is often imprecise, especially in crowded scenes. Therefore, *Label-2* is only used to adjust the question distribution (3–5 people: 60, 6–8 people: 60, 9+: 54). The ground truth number is manually annotated, and distractors are constructed based on this value. The distractors construction rule is to randomly select three different numbers within a range of up to 4 from the ground truth number, excluding the ground truth itself.

**Appearance Time Detection**: First, select videos based on the following criteria: the video duration must exceed 7 seconds, and the target individual's presence should account for between one-third and two-thirds of the total video length (calculated as the ratio of the human track frames to the total frames), primarily using *Label-1*. Then the frame range from *Label-1* is used to determine the target individual's appearance time range (format both ends as integers), which serves as the ground truth for generating questions about this person.

To obtain a detailed and accurate description of the individual, the same method as in the Human-to-Text task is used to generate the task-specific caption for the target. Using the description, questions are constructed in a template-based manner, as shown in Figure 14.

For distractor construction, three random time intervals are generated near the ground truth time interval, ensuring that their overlap with the ground truth interval does not exceed 4 seconds. This ensures the distractors do not cause confusion when selecting the correct answer.

Note that in this task, videos with bounding boxes are only used during the automatic description generation by Video-MLLMs and for manual verification. In the final version of the questions, the videos do not include bounding boxes.

**Action at Specified Time** and **Time of Specific Action** tasks rely on manual annotation, as attempts with various open-source models revealed their inability to accurately identify the timing of specified actions. For manual annotation, annotators are required to watch the videos and observe whether the highlighted individual performs any distinct short-term actions (quickly completed actions or "the start of an action", but not continuous states). They should record the action and its starting time. The videos and target individuals are consistent with those in Behavior Temporal Analysis task. Based on the specific action–time pairs provided by the annotators, two types of action-time-related questions are constructed.

| Labels | Human Emotion Perception | | | | Person Recognition | | | | Human Behavior Analysis | | | | Speech-Visual Alignment | | | |
|---|---|---|---|---|---|---|---|---|---|---|---|---|---|---|---|---|
| | ER | ETA | AR | EIC | T2H | H2T | HC | ATD | BTA | BCA | AST | TSA | AVSM | ASD | AVAD | SCM |
| *Label-1* | ✓ | ✓ | ✓ | ✓ | ✓ | ✓ | | ✓ | ✓ | ✓ | ✓ | ✓ | ✓ | ✓ | | |
| *Label-2* | | | | | ✓ | ✓ | ✓ | | | | | | ✓ | ✓ | ✓ | ✓ |
| *Label-3* | | | | | | | | | | | | | ✓ | | | |
| *Label-4* | ✓ | ✓ | ✓ | | ✓ | ✓ | | ✓ | ✓ | ✓ | ✓ | | | | | |
| *Label-5* | ✓ | ✓ | | | | | | | | | | | | | | |
| *Label-6* | | | | | | | | | | | | | ✓ | ✓ | ✓ | ✓ |
| *Label-7* | | | | | | | | | | | | | | ✓ | ✓ | ✓ |
| *Label-8* | | | | | | | | | | | | | | | | ✓ |
| *Label-10* | | | | | | | | | | | | | ✓ | | | |

Table 8: Annotation labels used in the construction process of 16 tasks

For the Action at Specified Time task, the question video consists of the highlighted target individuals with red bounding boxes. Only video samples where short-term actions are present are selected for question generation. The question template is shown in Figure 17. The ground truth is the specific action annotated for the individual. Distractors are generated by LLM from the task-specific captions in Behavior Temporal Analysis, with the following prompt:

*Please select and modify 3 actions from the list below to ensure that each action is significantly different from the target action ⟨ground truth action⟩. Here is the original action list: ⟨action_list⟩. Begin with "begin to .." for an action. If the number of actions is less than 3, generate one.*,

where the ⟨action_list⟩ is the sequence of actions generated by using LLM to summarize the task-oriented caption.

For the Time of Specific Action task, the question video is marked with target individual's bounding boxes. The question template is shown in Figure 18. For question samples with short-term actions, distractors are generated by selecting three numbers that are at least 3 seconds apart from the ground truth action time. For videos where a continuous action state is maintained throughout, the ground truth is set to "This is an ongoing action in the video." The distractors for such cases are fixed at 1s, 4s, 7s, and 10s. Note that to keep the options consistent, each question includes the option "This is an ongoing action in the video."

**Audio-Visual Speaker Matching**: First, select the appropriate question videos. The constraints mainly include video conditions and character conditions. The video conditions include: the audio label being "Speech", the number of people in the video is between 2 and 4, and the video duration is not less than 4 seconds; the character condition is: the frame coverage of the character must reach more than 67% of the video frame number. Further, the target person is selected as the ground truth according to the correlation between the age and gender attributes of the audio and the appearance of the person. Specifically, if the audio age belongs to "child", the only child is selected from the video as the target person, and the other characters are interference characters; If the audio age is "adult", the only adult with the same gender as the audio is selected from the video as the target person, and the other characters are interference characters. The age information is binary here because the audio age attribute is relatively vague. In addition, the gender characteristics of children's voices are not always distinguishable. Therefore, in order to enhance the optionality of the answer, the character types are divided into only three categories: male, female, and child. The question video uses *Label-1* to mark each optional person and capital letters as the option.

**Active Speaker Detection**: Suitable question videos are selected based on the criteria of having an audio label of "Speech", 2 to 4 people, and a duration of at least 4 seconds. The video must contain a single active speaker. *Label-1* is used to label all individuals, with the active speaker's label as the ground truth and others as distractors. Since the automated active speaker labels may not always be reliable, two additional options are included for each question to facilitate manual correction later: "There are more than one active speaker in the video." and "There is no active speaker in the video."

**Audio-Visual Alignment Detection**: Suitable question videos are selected based on the criteria of having fewer than 3 people, a duration of over 8 seconds, an audio type of "speech", and at least one active speaker. The video is then divided into three equal segments, with the left endpoint of each segment (rounded to an integer) used as potential options. One of these options is randomly chosen as the ground truth. The video is then modified by reversing the audio before the selected timestamp to create a "misaligned audio-visual" video.

**Speech Content Matching**: Videos are selected for question creation based on the following criteria: single-person scenes, duration greater than 5s, audio type "speech", and the person is an active speaker, the speech content being English with its sentence length greater than 35 characters. The ground truth is the automatic speech recognition result corresponding to *Label-8*. Distractors are generated using an LLM, which creates three different sentences with similar meaning and length to the ground truth as distractors.

In Table 8, we illustrate which labels from the Human-Centric Video Annotation Pipeline are used to construct each task.

## I.3 DETAILS OF HUMAN EFFORTS FOR HUMANVBENCH

We employed two professional full-time AI data annotators and invited two graduated-level volunteers to participate in the construction and evaluation of HUMANVBENCH. They collaborate to complete a series of tasks, with two main annotators participating in the full data of each task and two volunteers participating in the sampled data. Inconsistencies will be identified and resolved (for example, through discussion or majority voting to reach a consensus) to ensure high quality. The average annotation time per question is 3 minutes, totaling 10 workdays. Specifically, their tasks include the following four parts.

**Human Annotation on Generated QAs**: Tasks requiring human annotations to generate questions and options include Human Counting, Action at Specified Time, and Time of Specific Action. The latter two tasks can streamline annotation by annotating a single dataset containing "special short-term action & moment of occurrence". Therefore, in this step, each annotator was assigned to one annotation set. All other tasks were generated automatically, reducing the cost of human annotation.

**Manual Verification and Correction**: Except for the tasks that are already reliable enough, which include three tasks derived from the aforementioned human annotations, the Audio-Visual Alignment Detection and Speech Content Matching tasks, all other tasks require manual verification to ensure quality, following the process described in Section 3.3. During correction, low-quality samples (e.g., person transitions in human tracking, video freezing midway) are required to be flagged for removal.

**Cross-Verification**: After completing the above steps, we obtained 16 usable tasks for evaluation. To further ensure the high quality of the questions, we conducted cross-verification on 16 tasks except Emotion Recognition in Conversaton task to reduce the impact of personal biases and errors on the benchmark. Specifically, the tasks were cross-assigned to the two major annotators. For each task, the annotator in this step was ensured to be different from those responsible for the Manual Verification and Correction or Human Annotation. Annotators were first required to answer the multiple-choice questions. For disputed questions where answers were marked "incorrect", the new correct answer or option will be updated for this question, and these disputed questions were reassigned to another annotator for a second review. If errors persisted, both annotators discussed and agreed on a unified answer to serve as the final ground truth.

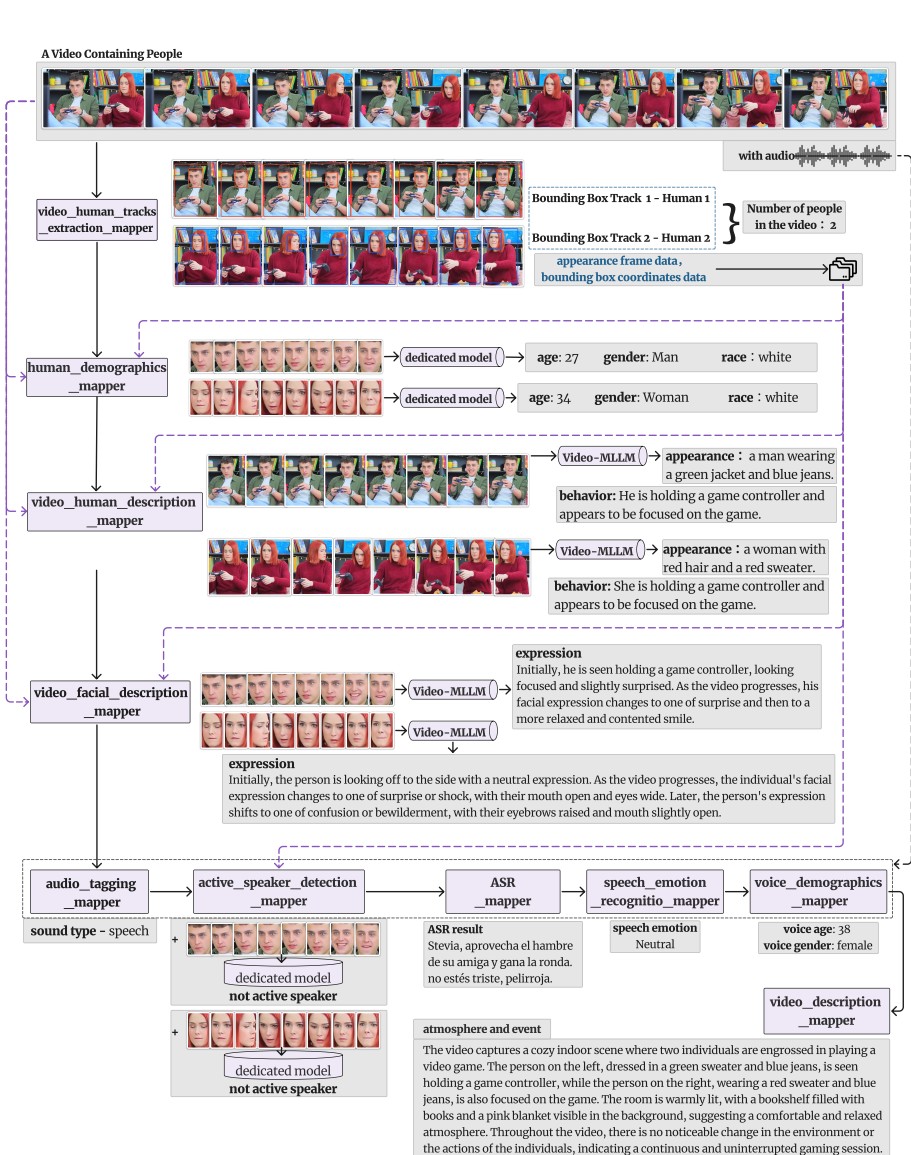

Figure 23: An example of using Human-Centric Annotation Pipeline for annotation.