# OpenReview forum: "HumanVBench: Probing Human-Centric Video Understanding in MLLMs with Automatically Synthesized Benchmarks"
_ICLR.cc/2026/Conference — ICLR 2026 Conference Withdrawn Submission_

### Official Review · Reviewer_H8bD · 2025-11-01

**Soundness:** 3
**Presentation:** 3
**Contribution:** 2
**Rating:** 6
**Confidence:** 4

**Summary:**

This paper introduces HUMANVBENCH, a new benchmark designed to evaluate the human-centric video understanding capabilities of Multimodal Large Language Models (MLLMs). The pipeline automatically generates video annotations and, most notably, uses model errors to create challenging and plausible multiple-choice distractors. The authors evaluated 27 MLLMs, revealing a significant performance gap between current models and humans, especially in nuanced emotion and speech-visual tasks.

**Strengths:**

- This work addresses a critical and timely problem. As MLLMs become more capable, the need for evaluation benchmarks that move beyond coarse-grained action recognition to probe subtle, human-centric understanding (like emotion, causality, and cross-modal coherence) is paramount.
- The 16-task benchmark is thorough and well-structured. It covers a wide range of capabilities, from perception (Human Counting ) to complex reasoning (Behavior Causality Analysis ) and critical cross-modal alignment (Audio-Visual Alignment Detection ). This multi-faceted approach allows for a detailed diagnosis of model strengths and weaknesses.

**Weaknesses:**

- The benchmark's "ground truth" is heavily reliant on the output of the Human-Centric Video Annotation Pipeline, which is composed of numerous other SOTA models. Although this is mitigated by a manual verification step, there is a risk that systematic biases or failure modes of the annotation models are propagated into the benchmark.
- While the combination, fine-grained nature, and synthesis method are novel, many of the individual tasks themselves are established problems in video understanding (e.g., Emotion Recognition, Human Counting, Active Speaker Detection). The paper does a good job of positioning itself against related work , but the primary innovation clearly lies in the benchmark generation and evaluation dimensions rather than the creation of entirely new task paradigms.

**Questions:**

Miss some good video llms works, e.g., Kimi-VL, Seed1.5-VL.

---

### Official Review · Reviewer_du2e · 2025-11-01

**Soundness:** 2
**Presentation:** 2
**Contribution:** 2
**Rating:** 6
**Confidence:** 3

**Summary:**

The authors introduce HumanVBench, a synthetic benchmark dataset specifically designed to evaluate how well MLLMs can understand videos centered around human activities and interactions. The dataset consists of synthetic video clips with accompanying text prompts that are crafted to probe various aspects of human-centric understanding, such as recognizing human actions, emotions, social interactions, and reasoning about human intentions.

**Strengths:**

- The paper introduces a novel benchmark dataset specifically tailored to evaluate the human-centric video understanding capabilities of Multimodal Large Language Models (MLLMs).
- The dataset is carefully designed with diverse scenarios to cover a wide range of human-centric tasks. The authors provide detailed descriptions of the data generation process, ensuring reproducibility and reliability.
- The paper is well-organized and clearly written.

**Weaknesses:**

- While synthetic data offers control over scenarios and variables, it may not fully capture the complexity and variability of real-world videos.
- The current metrics may not fully capture the depth of reasoning required for human-centric video understanding. For example, evaluating a model’s ability to infer intentions or predict future actions might require more nuanced metrics.

**Questions:**

Could the authors elaborate on how well the results obtained from the synthetic benchmark translate to real-world performance?

---

### Official Review · Reviewer_Lp2P · 2025-11-01

**Soundness:** 2
**Presentation:** 3
**Contribution:** 2
**Rating:** 4
**Confidence:** 3

**Summary:**

HUMANVBENCH presents a human-centric video understanding benchmark with 16 tasks and automated pipelines for annotation and QA generation.   The evaluation of 27 MLLMs highlights clear weaknesses in emotion perception and audio visual alignment.   The work has strong motivation and practical value, but data diversity, transparency, and deeper analysis are needed.   Overall, the benchmark is a useful contribution to human-centric multimodal evaluation.

**Strengths:**

1.   The paper targets an important gap in video MLLM evaluation.   Existing benchmarks overlook human-centric reasoning such as fine-grained emotion, social interaction, speaker grounding, and audio-visual alignment.   HUMANVBENCH provides 16 well-defined tasks that clearly separate internal human states from external behaviors.

2.   The benchmark construction is innovative and scalable.   The dual automatic pipelines and use of 20+ operators enable fine-grained annotation and generate realistic distractors while reducing human labor.   The modular design can transfer to other domains, suggesting strong extensibility.

3.   The empirical study is comprehensive and convincing.   Twenty-seven video MLLMs are evaluated with human and random baselines.   The results consistently reveal weaknesses in emotion understanding and cross-modal synchronization.   Annotation quality is validated and ablation studies improve credibility.

**Weaknesses:**

1.    Data diversity is limited.    Pexels and MF2 provide controlled settings but lack challenging real-world scenarios such as occlusion, low-resource speech, or casual social interaction.    The benchmark may not fully reflect in-the-wild performance.

2. Although the paper tests adding timestamps to models that do not natively support temporal modeling, these models were never trained to align events with time, so the improvement from timestamps may be inherently limited. This makes the experiment less conclusive, because it cannot fairly compare against models built with temporal reasoning mechanisms by design.

3.    Several implementation details are missing.    Operator versions, parameter settings, prompt templates, and hyperparameters are not fully documented, reducing reproducibility.

4.    Related work coverage is incomplete.    Recent human-centric benchmarks and adversarial QA generation frameworks are not sufficiently compared, making innovation boundaries unclear.

**Questions:**

1.  Is the wav2vec2-based audio attribute labeling accurate on your data, and how is label noise handled?
2.  Why is there no few-shot or low-resource evaluation, given its importance in practical deployment?
3.  Has potential data leakage been checked for commercial models that perform exceptionally well on MF2?
4.  How much adaptation effort is required to transfer the pipeline to non-human domains, and how is distractor quality evaluated?
5.  How are privacy risks addressed for facial and speech data, and is anonymization applied?

---

### Official Review · Reviewer_SMgx · 2025-11-01

**Soundness:** 3
**Presentation:** 3
**Contribution:** 2
**Rating:** 2
**Confidence:** 4

**Summary:**

This paper presents HUMANVBENCH, a benchmark for evaluating human-centric video understanding in multimodal large language models (MLLMs). It focuses on assessing fine-grained abilities such as emotion perception, behavior analysis, and audio-visual alignment, which are often neglected in existing benchmarks.The benchmark includes 16 tasks covering multiple aspects of human understanding and is used to evaluate 27 leading MLLMs.Results show that current models, including strong proprietary ones like Gemini-2.5-Pro and GPT-5, still fall short of human-level comprehension, especially in emotional and cross-modal reasoning. Overall, HUMANVBENCH offers a scalable and fine-grained evaluation framework that advances the study of human-centric intelligence in MLLMs, though some methodological details could be clarified further.

**Strengths:**

1.The paper investigates a meaningful research problem, focusing on the fine-grained evaluation of MLLMs in understanding human emotions, behaviors, and audio-visual alignment.
2.It provides a detailed introduction to the automated annotation and QA generation framework.
3.The benchmark is comprehensive, covering 16 tasks and evaluating 27 MLLMs, offering systematic and broad assessment.

**Weaknesses:**

1.The paper introduces an automated annotation and QA generation pipeline, but its scalability and effectiveness for large-scale training data or model performance improvement are not empirically demonstrated.
2.The related work section discusses several existing benchmarks, yet the comparison remains qualitative and lacks a systematic or quantitative analysis to clearly position HUMANVBENCH.
3.The discussion of model failure cases is brief and lacks deeper insights into the underlying causes, reducing the benchmark’s diagnostic and interpretive value.

**Questions:**

Please refer to the reference.

---

### Note · Authors · 2025-11-14

I have read and agree with the venue's withdrawal policy on behalf of myself and my co-authors.